# Assessing the benefits of approximately exact step sizes for Picard and Newton solver in simulating ice flow (FEniCS-full-Stokes v.1.3.2)

**Niko Schmidt**[1]**, Angelika Humbert**[2,3]**, and Thomas Slawig**[1,4]

[1]Department of Computer Science, Kiel University, Kiel, Germany
[2]Alfred-Wegener-Institut Helmholtz-Zentrum für Polar- und Meeresforschung, Bremerhaven, Bremen, Germany
[3]Faculty of Geosciences, University of Bremen, Bremen, Germany
[4]Kiel Marine Science (KMS) – Centre for Interdisciplinary Marine Science, Kiel, Germany

**Correspondence:** Niko Schmidt (n_f_schmidt@yahoo.de)

**Abstract.** Solving the momentum balance is the computationally expensive part of simulating the evolution of ice sheets. The momentum balance is described by the nonlinear full-Stokes equations, which are solved iteratively. We use the Picard iteration and Newton's method combined with Armijo step sizes and approximately exact step sizes, respectively, to solve these equations. The Picard iteration uses either no step size control or the approximately exact step sizes. We compare the variants of Newton's method and the Picard iteration in benchmark experiments, called ISMIP-HOM experiments A, B, E1, and E2. The ISMIP-HOM experiments consist of a more realistic domain and are designed to test the quality of ice models. For an even more realistic test case, we simulate the experiments E1 and E2 with a time-dependent surface. We obtain that approximately exact step sizes greatly reduce the necessary number of iterations for the Picard iteration and Newton's method with nearly no increase in the computation time for each iteration.

## 1 Introduction

Simulating the evolution of the ice sheets in Greenland and Antarctica in adequate physics and resolution is a challenging task. The dynamics of ice sheets is described as a fluid mechanical problem with the momentum balance reduced to a Stokes problem as acceleration and Coriolis forces are negligible. In the past, computational constraints led to the reduction of the problem by approximating the momentum balance. If a sufficiently large spatial resolution cannot be chosen, the benefit from solving the Stokes problem is lost. Consequently, in practical terms, Stokes models lead to large problems, and thus efficient solvers are inevitable. This is what this study focuses on.

The full-Stokes equations are nonlinear partial differential equations described as shear thinning, which means that the viscosity depends nonlinearly on the symmetric gradient. More precisely, we consider the stationary variant of these equations in the variational formulation. In ice models, the stationary equations are solved to calculate the velocity field. The velocity field is then used to calculate the new shape of the glacier. The variational formulation is needed to calculate a solution with finite elements. A common method of calculating the solution of these equations is the Picard iteration (see Colinge and Rappaz, 1999). The Picard iteration fixes the nonlinear viscosity, calculates a new velocity, and updates the viscosity.

The Picard iteration is, for example, used in the ice models ISSM (Larour et al., 2012) and FELIX-S (Leng et al., 2012). Elmer/Ice (Gagliardini et al., 2013) uses the Picard iteration for the first few iterations and the Newton method for the last iterations. This approach was also extended to a nonlinear friction law. However, Elmer/Ice does not use a step size control. Thereby, Newton's method does not converge from every initial point (Gagliardini et al., 2013). The ice models Elmer/Ice and FELIX-S are compared in Zhang et al. (2017). For some glacier simulations, COMSOL multiphysics is used

(see Rückamp et al., 2022). Newton's method in COMSOL multiphysics switches between choosing the negative gradient and Newton directions. Additionally, a trust-region method is used to determine the step sizes. The trust-region radius is the maximum step size that one would trust the step size to be so it is a good choice (for more mathematical details, see Sect. 6.4.2 in Dennis and Schnabel (1996)). The trust-region method in COMSOL multiphysics uses the residual norm (see Sect. "The Fully Coupled Attribute and the Double Dogleg Method" in COM, 2018). Instead, we will discuss another approach, which is algorithmically simpler and uses the problem-specific information of a convex function. Additionally, the convex function allows us to use different step size methods.

We employ Newton's method by formulating the variational formulation as a root problem. If we start near the solution, Newton's method is superlinear convergent (Hinze et al., 2009). Thus, the error between the approximation and the real solution reduces faster than the linear convergent Picard iteration (see Fraters et al., 2019). However, starting with an unsuitable initial velocity field for Newton's method could lead to a diverging velocity. A step size control guarantees convergence from every initial guess. This step size control is constructed by defining a function that we want to minimize. One variant is presented in Fraters et al. (2019). We consider another approach that only needs to calculate integrals. It allows us to use two different step size controls. Newton's method with one of these step sizes converges from every initial guess to the solution (Schmidt, 2023). Additionally, we employ approximately exact step sizes for the Picard iteration to provide a possibility of reducing the necessary number of iterations without implementing Newton's method. The exact step sizes are the solution of a one-dimensional minimization problem. As we can only approximate these step sizes arbitrarily precisely, we called them approximately exact and call them exact step sizes in this paper for brevity.

The computation of the step size is computationally cheap compared to solving the linear systems of equations in each iteration. The work of Habbal et al. (2017) considers different solvers to reduce the simulation time for solving the system of linear equations. Nonetheless, for all solvers, the system of linear equations is still the main computational effort. Our step size control reduces the computation time by reducing the necessary number of iterations.

As a test case, we use the ISMIP-HOM experiments A, B, E1, and E2. These experiments are designed to test the quality of glaciological models. They reflect a large domain of the glaciers, a large aspect ratio, and a sinusoidal bedrock and the Glacier d'Arolla, respectively. We simulate the Glacier d'Arolla experiments E1 and E2 also as time-dependent.

The paper has the following structure: in Sect. 2, we introduce the equations in the variational formulation and the Picard iteration. In the subsequent section, we formulate Newton's method. In Sect. 4, we introduce the new idea, the step

size control that decreases the number of iterations and verifies convergence from every initial guess. In Sect. 5, we discuss the stationary ISMIP-HOM experiments A and B and compare them with the results in Pattyn et al. (2008). In Sect. 6, we solve instationary problems with and without sliding derived from the ISMIP-HOM experiments E1 and E2. Finally, we give a summary in Sect. 7 and an outlook in Sect. 8.

## 2  The full-Stokes equations as a root problem

Let $\Omega \subseteq \mathbb{R}^N$, with $N \in \{2, 3\}$. For describing the movement of ice, we need the second-order tensor $\boldsymbol{\sigma}$, the density $\rho$, and the gravitational acceleration $\boldsymbol{g}$. These quantities describe the full-Stokes equations, the most complex equations for simulating ice, by

$$\begin{aligned} -\mathrm{div}\,\boldsymbol{\sigma} &= -\rho\boldsymbol{g}, \\ \mathrm{div}\,\boldsymbol{v} &= 0, \end{aligned} \tag{1}$$

on the domain $\Omega$. We describe the stress tensor, $\boldsymbol{\sigma}$, with the pressure, $p$; the identity tensor (matrix), $\mathbf{I}$; the symmetric gradient, $D$; the velocity, $\boldsymbol{v}$; and the viscosity, $\mu$, by $\boldsymbol{\sigma} := p\mathbf{I} - \mu D\boldsymbol{v}$. We define the nonlinear viscosity, $\mu$, as

$$\begin{aligned} \mu &= B\big(|D\boldsymbol{v}|^2 + \delta^2\big)^{\frac{1-n}{2n}}, \\ (D\boldsymbol{v})_{ij} &= \frac{1}{2}\left(\frac{\partial v_i}{\partial x_j} + \frac{\partial v_j}{\partial x_i}\right), \\ |D\boldsymbol{v}|^2 &:= D\boldsymbol{v} : D\boldsymbol{v} := \sum_{i,j=1}^{N}|(D\boldsymbol{v})_{ij}|^2, \end{aligned} \tag{2}$$

with $n \in (1, \infty)$ and $B = B(x_1, x_2, x_3)$, $\delta > 0$. The constant $\delta > 0$ guarantees $\mu < \infty$. We choose $n = 3$ for the experiments as in Pattyn et al. (2008). The boundary consists of the bedrock, $\Gamma_b$; the surface, $\Gamma_s$; and the lateral boundary, $\Gamma_\ell$. Our boundary conditions are

$$\begin{aligned} \boldsymbol{v} &= \boldsymbol{0} &&\text{on } \Gamma_b \cup \Gamma_\ell, \\ \boldsymbol{\sigma} \cdot \boldsymbol{n} &= \boldsymbol{0} &&\text{on } \Gamma_s, \end{aligned} \tag{3}$$

with the outer normal vector $\boldsymbol{n}$. Here, $\boldsymbol{\sigma} \cdot \boldsymbol{n}$ is the inner tensor-product (matrix–vector multiplication).

In Sect. 6, we also consider the sliding boundary condition $\boldsymbol{v} \cdot \boldsymbol{n} = 0$. Due to zero values of the friction coefficient in this experiment, only the spaces for the variational formulation change by including $\boldsymbol{v} \cdot \boldsymbol{n} = 0$. Thus, for simplicity, we neglect this additional boundary condition in our upcoming derivation.

We derive the variational formulation in infinite dimensions because we can implement it directly in FEniCS (see Logg et al., 2012). We determine the variational formulation by multiplying with test functions and using partial integration, and, second, we explain the function spaces used. We define an operator $G : H \times L \to H^* \times L^*$ by

$$\langle G(\boldsymbol{v}, p), (\boldsymbol{\phi}, \psi) \rangle = \int_\Omega B\big(|D\boldsymbol{v}|^2 + \delta^2\big)^{\frac{1-n}{2n}} D\boldsymbol{v} : \nabla\boldsymbol{\phi}\,\mathrm{d}x$$

$$+ \mu_0 \int_\Omega \nabla\boldsymbol{v} : \nabla\boldsymbol{\phi}\,\mathrm{d}x - \int_\Omega p\,\mathrm{div}\boldsymbol{\phi}\,\mathrm{d}x - \int_\Omega \mathrm{div}\boldsymbol{v}\,\psi\,\mathrm{d}x$$

$$+ \int_\Omega \rho\boldsymbol{g}\cdot\boldsymbol{\phi}\,\mathrm{d}x, \tag{4}$$

where $\boldsymbol{v} \in H$ and $p \in L$ are the solution and $\boldsymbol{\phi} \in H$ and $\psi \in L$ are test functions. The angular brackets on the left-hand side of the equation are used because, formally, we have a function that maps to the dual space. The dual space is denoted by the star after the space, e.g., $H^*$ and $L^*$. The solution of the full-Stokes equations is $(\boldsymbol{v}, p) \in H \times L$, with

$$\langle G(\boldsymbol{v}, p), (\boldsymbol{\phi}, \psi) \rangle = 0 \quad \text{for all } (\boldsymbol{\phi}, \psi) \in H \times L. \tag{5}$$

We added the diffusive term $\mu_0 > 0$ to get a well-posed directional derivative and a well-posed Picard iteration (for details, see Appendix A1). Additionally, we are now in a Hilbert space formulation and set $H := \{\boldsymbol{v} \in H^1(\Omega)^N;\ \boldsymbol{v}|_{\Gamma_b \cup \Gamma_\ell} = \boldsymbol{0}\}$, where $H^1(\Omega)^N$ is the space of vector-valued square integrable functions with a square integrable derivative. We set $L := \{p \in L^2(\Omega); \int_\Omega p\,\mathrm{d}x = 0\}$ for the space of square integrable functions with a zero integral. There exists a unique solution to that problem for $\mu_0 > 0$ (see Schmidt, 2023). Nonetheless, we perform all experiments with $\mu_0 = 0$.

We formulate the problem in infinite-dimensional spaces $H$ and $L$. In these infinite-dimensional spaces, mathematical convergence properties are independent of the mesh resolution and the finite elements used as long as the finite elements are a subspace of the infinite-dimensional spaces. Ice models often use finite elements. Moreover, the formulation in discretized spaces is identical; only the functions are from finite-dimensional spaces.

A common method to solve the variational formulation of the full-Stokes equations in glaciological models is the Picard iteration (see Algorithm 1). It is used in ISSM (see Larour et al., 2012), FELIX-S (see Leng et al., 2012). Elmer/Ice can use the Picard iteration and Newton's method, which we introduce in the next section. Elmer/Ice can use the Picard iteration to get near the solution and then can use Newton's method (see Gagliardini et al., 2013).

## 3 Newton's method

The Picard iteration converges slowly (see Fraters et al., 2019). Thus, it can be beneficial to consider faster-converging algorithms. Newton's method is often superlinear convergent, also in infinite dimensions (see Hinze et al., 2009). For Newton's method, the calculation of the derivative is necessary. Due to the variational formulation, we can

---

**Algorithm 1** Picard iteration.

1: Let $\boldsymbol{v}_0 \in H$ and $p_0 \in L$ be given.
2: **for** $k = 0, 1, \ldots$ **do**
3:     Calculate $\boldsymbol{v}_{k+1} \in H$ and $p_{k+1} \in L$ with

$$\int_\Omega B\big(|D\boldsymbol{v}_k|^2 + \delta^2\big)^{\frac{1-n}{2n}} D\boldsymbol{v}_{k+1} : \nabla\boldsymbol{\phi}\,\mathrm{d}x$$

$$+ \mu_0 \int_\Omega \nabla\boldsymbol{v}_{k+1} : \nabla\boldsymbol{\phi}\,\mathrm{d}x - \int_\Omega p_{k+1}\mathrm{div}\boldsymbol{\phi}\,\mathrm{d}x$$

$$- \int_\Omega \mathrm{div}\boldsymbol{v}_{k+1}\psi\,\mathrm{d}x = -\int_\Omega \rho\boldsymbol{g}\cdot\boldsymbol{\phi}\,\mathrm{d}x$$

    for all $\phi \in H$ and $\psi \in L$.
4: **end for**

---

only express the derivative of $G$ in terms of the direction and the test functions. The derivative of $G$ in $(\boldsymbol{v}, p)$ in direction $(\boldsymbol{w}, q)$ is

$$\langle G'(\boldsymbol{v}, p)(\boldsymbol{w}, q), (\boldsymbol{\phi}, \psi) \rangle$$

$$= \int_\Omega \frac{1-n}{n} B\big(|D\boldsymbol{v}|^2 + \delta^2\big)^{\frac{1-3n}{2n}} (D\boldsymbol{v} : D\boldsymbol{w})\,(D\boldsymbol{v} : \nabla\boldsymbol{\phi})\,\mathrm{d}x$$

$$+ \mu_0 \int_\Omega \nabla\boldsymbol{w} : \nabla\boldsymbol{\phi}\,\mathrm{d}x$$

$$+ \int_\Omega B\big(|D\boldsymbol{v}|^2 + \delta^2\big)^{\frac{1-n}{2n}} D\boldsymbol{w} : \nabla\boldsymbol{\phi}\,\mathrm{d}x$$

$$- \int_\Omega q\,\mathrm{div}\boldsymbol{\phi}\,\mathrm{d}x - \int_\Omega \mathrm{div}\boldsymbol{w}\,\psi\,\mathrm{d}x. \tag{6}$$

A mathematical proof that $G$ is differentiable in all directions $(\boldsymbol{w}, q)$ is presented in Schmidt (2023). A more detailed deduction of the derivative is in Sect. A2.

Newton's method can solve the full-Stokes equations by Algorithm 2. Because Newton's method is only locally convergent, we use a step size control in Algorithm 2. We explain the step size control in the next section.

---

**Algorithm 2** Globalized Newton's method.

1: Let $(\boldsymbol{v}_0, p_0)$ be given.
2: **for** $k = 0, 1, \ldots$ **do**
3:     Calculate $(\boldsymbol{w}_k, q_k)$ with

$$\langle G'(\boldsymbol{v}_k, p_k)(\boldsymbol{w}_k, q_k), (\boldsymbol{\phi}, \psi) \rangle =$$
$$- \langle G(\boldsymbol{v}_k, p_k), (\boldsymbol{\phi}, \psi) \rangle \text{ for all } \boldsymbol{\phi} \in H, \psi \in L.$$

4:     Set $\boldsymbol{v}_{k+1} := \boldsymbol{v}_k + \alpha_k \boldsymbol{w}_k$ and $p_{k+1} := p_k + \alpha_k q_k$ with a suitable $\alpha_k > 0$.
5: **end for**

## 4 Step size control

In this section, we derive a global convergent Newton method using a step size control; we have the current velocity field $\boldsymbol{v}$ and the direction $\boldsymbol{w}$. Instead of setting our new field, $\tilde{\boldsymbol{v}} := \boldsymbol{v} + \boldsymbol{w}$, we choose $\alpha > 0$, with $\tilde{\boldsymbol{v}} := \boldsymbol{v} + \alpha \boldsymbol{w}$. We want an algorithm for choosing this $\alpha$. Classical approaches for determining the step size $\alpha$ check if the norm $\|G(\boldsymbol{v}_{k+1}, p_{k+1})\|$ reduces enough compared to $\|G(\boldsymbol{v}_k, p_k)\|$. What enough reduction means is discussed in, for example, Hinze et al. (2009). However, we use an alternative approach. Solving $G(\boldsymbol{v}, p) = 0$ is equivalent to minimizing $J : H \times L \to \mathbb{R}$ as in Eq. 7 (see Schmidt, 2023).

$$
J(\boldsymbol{v}, p) = \int_\Omega \frac{n}{1+n} B\big(|D\boldsymbol{v}|^2 + \delta^2\big)^{\frac{1+n}{2n}} \, \mathrm{d}x
$$
$$
+ \frac{\mu_0}{2} \int_\Omega |\nabla \boldsymbol{v}|^2 \, \mathrm{d}x + \int_\Omega \rho \boldsymbol{g} \cdot \boldsymbol{v} \, \mathrm{d}x - \int_\Omega p \, \mathrm{div} \boldsymbol{v} \, \mathrm{d}x. \quad (7)
$$

We need the last summand because the time-dependent experiments lead to initial guesses for the velocity field that are not divergence-free; we start with a divergence-free initial guess, calculate the velocity field, and use the velocity field to calculate the new domain. (We explain the calculation of the new domain in Sect. 6.2.) The grid points with the velocity information are moved correspondingly to fit the new domain. On this new domain, our old velocity field is our initial guess and slightly not divergence-free. (Despite $\mathrm{div} \boldsymbol{v}$ being near 0, the step size control did not work without the last summand in Eq. 7 for time-dependent problems.)

The convex functions were also used in Hirn (2013) for $\mu_0 = 0$ with Dirichlet boundary conditions, and Chen et al. (2013) for $\delta = 0$ and $\mu_0 = 0$, with more realistic boundary conditions. The equivalence between minimizing this convex function and solving the full-Stokes equations is clear because the minimizer of the function and the root of the derivative are at the same point for strict convex functions.

A classical approach to determine a suitable step size $\alpha$ is the use of an Armijo step size as in Hinze et al. (2009) (see Algorithm 3).

---

**Algorithm 3** Armijo step size.

---

1: Let $\gamma \in (0, 1/2)$.
2: **for** $i = 0, 1, \ldots$ **do**
3:     Set $\alpha := 1$.
4:     **while** $J(\boldsymbol{v} + \alpha \boldsymbol{w}, p + \alpha q) - J(\boldsymbol{v}, p) > \alpha \gamma J'(\boldsymbol{v}, p)(\boldsymbol{w}, q)$ **do**
5:         Set $\alpha := 0.5\alpha$.
6:     **end while**
7: **end for**
8: **return** $\alpha$

---

We describe the idea of Armijo step sizes: for a function, the negative gradient is the direction of the steepest descent. To find the minimum, we need enough reduction compared to the steepest descent multiplied with the step size and a factor between 0 and 1 as we expect less reduction than the steepest descent. This condition is stated in line 4 of Algorithm 3. Line 3 guarantees that the step size is not too small. For Newton's method, one chooses $\alpha := 1$ (see Nocedal and Wright, 2006, Algorithm 3.1). Newton's method converges fast for $\gamma \in (0, 1/2)$ (see Nocedal and Wright, 2006, Theorem 3.6). Typically, $\gamma := 10^{-4}$ is chosen (see Nocedal and Wright, 2006). However, we choose $\gamma := 10^{-10}$ as $10^{-4}$ was to strict. We only need a direction that reduces the function value to use Armijo step sizes.

However, we can exploit the convexity of $J$ for constructing other step sizes; we define the auxiliary function $J_k$:

$$
J_k(\alpha) := J(\boldsymbol{v}_k + \alpha \boldsymbol{w}_k, p_k + \alpha q_k). \quad (8)
$$

The function $J_k$ is strictly convex for $\boldsymbol{w}_k \neq 0$ because $J$ is strictly convex with respect to the first argument and convex with respect to the second argument. Thus, the following equation:

$$
J_k'(\alpha) = J'(\boldsymbol{v}_k + \alpha \boldsymbol{w}_k, p_k + \alpha q_k)(\boldsymbol{w}_k, q_k) \quad (9)
$$

is negative for $\alpha = 0$ and positive for a big enough $\alpha$ because of the strict convexity of $J_k$ for $\boldsymbol{w}_k \neq 0$. As $J_k'$ is continuous, a simple bisection (see Algorithm 4) calculates the minimum of $J_k$. In practice, we approximate the exact step size arbitrarily precisely. Thus, we denote the approximate exact step size as the exact step size. Exact step sizes have the advantage of allowing us to really calculate the minimum in a direction instead of just noting some reduction. Nonetheless, the exact step size is only rarely used in practice. One needs more conditions on the problem, which is the strict convexity of $J_k$ for $\boldsymbol{w}_k \neq 0$ here. We could not find the minimum of

$$
\|G(\boldsymbol{v}_k + \alpha \boldsymbol{w}_k, p_k + \alpha q_k)\|
$$

for the direction $(\boldsymbol{w}_k, q_k)$ because we have no information on where the minimum is.

For simplicity, we choose 25 iterations to approximate the exact step size. As we expect step sizes of length 1 to often be a good choice, we choose $a := 0$ for the minimum step length and $b := 4$ for the maximum step length in our implementation. We then obtain an accuracy of $4/2^{25} \approx 10^{-7}$ for the step size $\alpha$. The calculation of $\alpha$ is computationally not expensive.

We modify the Picard iteration (see Algorithm 1) by a step size control; we set

$$
\tilde{\boldsymbol{v}}_{k+1} := (1 - \alpha_k) \boldsymbol{v}_k + \alpha_k \boldsymbol{v}_{k+1},
$$
$$
\tilde{p}_{k+1} := (1 - \alpha_k) p_k + \alpha_k p_{k+1} \quad (10)
$$

and choose $\alpha_k$ as $\alpha$ in Algorithm 4.

## 5 Stationary experiments

We analyze the four algorithms we introduced: the Picard iteration without a step size control as a reference, exact

---

**Algorithm 4** Exact step size.

---

1: Set $a, b \in [0, \infty)$ with $a < b$.
2: **for** $i = 0, 1, \ldots$ **do**
3:     **if** $J_k'((a+b)/2) > 0$ **then**
4:         Set $b := (a+b)/2$.
5:     **else**
6:         Set $a := (a+b)/2$.
7:     **end if**
8: **end for**
9: **return** $\alpha := (a+b)/2$

---

step sizes for the Picard iteration and Newton's method, and Armijo step sizes for Newton's method. We implemented all these algorithms in FEniCS version 2019.1.0 (see Logg et al., 2012). FEniCS is a library that allows for the implementation of variational formulations easily. Hence, it allows for fast testing of algorithms without implementing them in complex codes. We determine the performance of these algorithms by comparing each iteration step with a reference solution for the experiments ISMIP-HOM A and B (see Pattyn et al., 2008). The reference solution is calculated with 80 prescribed Picard iterations without a step size control. We choose 80 iterations as this method converges slowly and the reference solution should be more accurate than the solutions calculated by the compared methods. In Pattyn et al. (2008), the authors described the ISMIP-HOM experiments to analyze the quality of ice models. Moreover, they compared simulation results.

We set the physical variables according to Pattyn et al. (2008): $B := 0.5 \times (10^{-16})^{-1/3}$ $(Pa)^{-3} a^{-1}$, $\rho := 910 \, kg \, m^{-3}$, and $g := (0, 9.81) \, m \, s^{-2}$.

We set the constant $\delta := 10^{-12} \, a^{-1}$ and $\mu_0 := 0 \, kg \, a \, m^{-1} \, s^{-2}$. We derive the unit for $\mu_0$ by $[\mu_0 |\nabla v|^2] = [\rho g \cdot v]$. In the experimental design, the nonlinear term is $2B(0.5|Dv|^2 + \delta^2)^{(1-n)/(2n)}$ instead of $B(|Dv|^2 + \delta^2)^{(1-n)/(2n)}$ (see Pattyn et al., 2008). We choose the constant $\delta$ such that $\delta$ is smaller than the typical magnitude of $Dv$, $3 \times 10^{-4}$ and $3 \, a^{-1}$, multiplied by the machine precision, eps:

$$\delta < \text{eps} \sqrt{0.5}|Dv| \tag{11}$$

for typical values of $|Dv|$. We defined the infinite-dimensional algorithms for $\mu_0 > 0$ as they are not well posed for $\mu_0 = 0$. However, we perform all simulations with $\mu_0 = 0$ because the additional diffusion term is not used in ice models and the diffusion term is not needed in the finite-dimensional finite-element spaces (Hirn, 2013). In all experiments, we calculate the initial velocity by replacing $(0.5|Dv|^2 + \delta^2)^{(1-n)/(2n)}$ with $10^6$ and solving this linear problem. As starting with $|Dv| = 3 \times 10^{-4} \, a^{-1}$ leads to $(0.5|Dv|^2 + \delta^2)^{(1-n)/(2n)} \approx 281 \, a^{2/3}$ and starting with a constant velocity field leads to $(0.5|Dv|^2 + \delta^2)^{(1-n)/(2n)} = 10^8 \, a^{2/3}$, we chose $10^6$ between both values.

## 5.1 The original experiment ISMIP-HOM B

In this subsection, we introduce details from Pattyn et al. (2008) that are specific to the experiment ISMIP-HOM B. This experiment has a domain with a sinusoidal, slightly tilted (0.5°) bottom. The boundaries at the left and right are vertical, and the boundary at the top has a linear slope of 0.5°. Furthermore, periodic boundary conditions are used at $\Gamma_\ell$. The experiment prescribes Dirichlet zero boundary conditions, $v = 0$ on $\Gamma_b$ and $\sigma \cdot n = 0$ on $\Gamma_s$.

The length $L := 5 \, km$ is the horizontal extent. The angle $\beta := 0.5°$ describes a slight decline at the surface and the bottom by

$$z_s(x) = -x \tan(\beta),$$
$$z_b(x) = z_s(x) - 1000 + 500 \cdot \sin(\omega x), \tag{12}$$

with $\omega := 2\pi/L$.

## 5.2 Modifications to the experiment ISMIP-HOM B

Formulating the convex function $J$ (see Eq. (7)) that corresponds to periodic boundary conditions is complicated. Thus, we use the alternative introduced in the supplement of Pattyn et al. (2008) by copying the glacier to the right and the left. We have three copies to the right and the left (see Fig. 1). At the lateral boundary, $\Gamma_\ell$, we impose Dirichlet zero boundary conditions. Also, the resolution at the outer copies is lower than for the original domain. This reduces the number of elements for the two-dimensional experiment by 30 % and in three dimensions by 51 %. Nevertheless, the three-dimensional experiment was performed on a high-performance computer. In two dimensions, the local refinement has no relevant impact on the solution. Also, one can simulate the two-dimensional experiment on a laptop.

Instead of the slope, we rotate the gravity. Thus, we should rotate the lateral boundary, $\Gamma_\ell$ of the domain. We neglect this and stick to vertical boundaries on the left and the right.

## 5.3 Results for experiment ISMIP-HOM B

Our velocity fields at the surface (see Fig. 2) are close to the mean of the full-Stokes simulations in Pattyn et al. (2008).

Also, all our methods produce very similar velocity fields at the surface, as displayed in Fig. 3. Next, we compare how many iterations are necessary to reduce the relative difference and relative local difference compared to the reference solution. We calculate the relative difference and the relative local difference for a velocity $v$ and the reference solution $v_{ref}$ by

$$\sqrt{\frac{\int_\Omega |v - v_{ref}|^2 \, dx}{\int_\Omega |v_{ref}|^2 \, dx}} \text{ and } \sqrt{\int_\Omega \frac{|v - v_{ref}|^2}{\max(|v_{ref}|^2, c^2)} \, dx}, \tag{13}$$

with $c = 1 \, mm \, a^{-1}$. Then, $c$ is much smaller than $10 \, m \, a^{-1}$, and below this velocity speed, slight differences are seen as

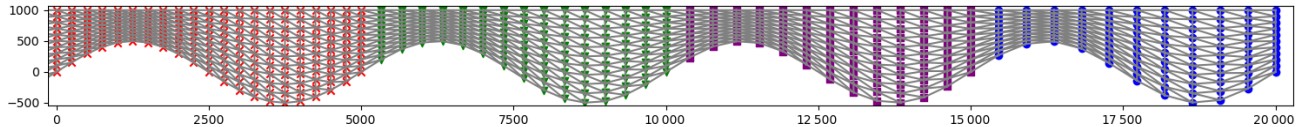

**Figure 1.** The domain, with a grid with red dots and three copies to the right with green, purple, and blue dots.

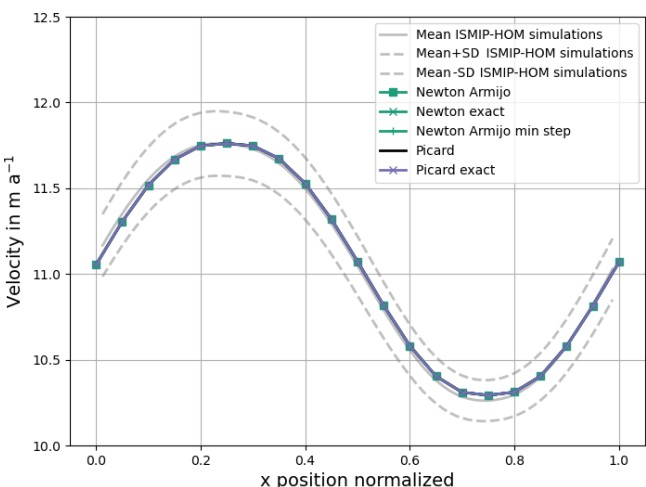

**Figure 2.** Simulated surface velocity for different solvers for ISMIP-HOM B. All our calculated velocity fields overlap with each other. The mean and the standard deviation (SD) from Pattyn et al. (2008) with nine models are plotted in grey. The mean and standard deviations have no values at $x = 0$ and $x = 1$ due to missing values.

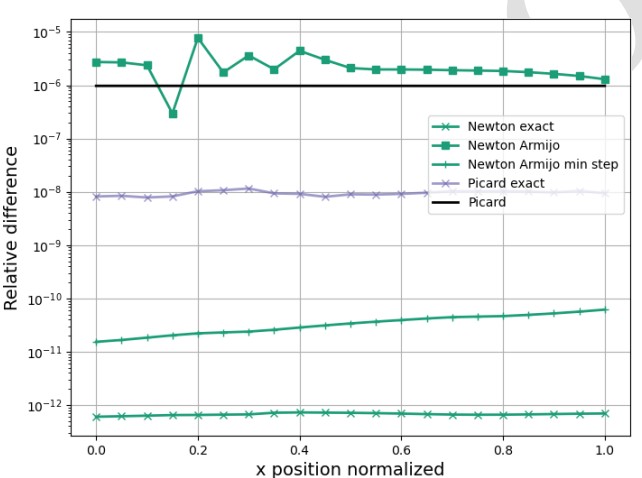

**Figure 3.** Relative difference in $|\boldsymbol{v} - \boldsymbol{v}_{\mathrm{ref}}|/|\boldsymbol{v}_{\mathrm{ref}}|$ for each grid point at the surface. The reference solution is the solution from 80 Picard iterations without a step size control.

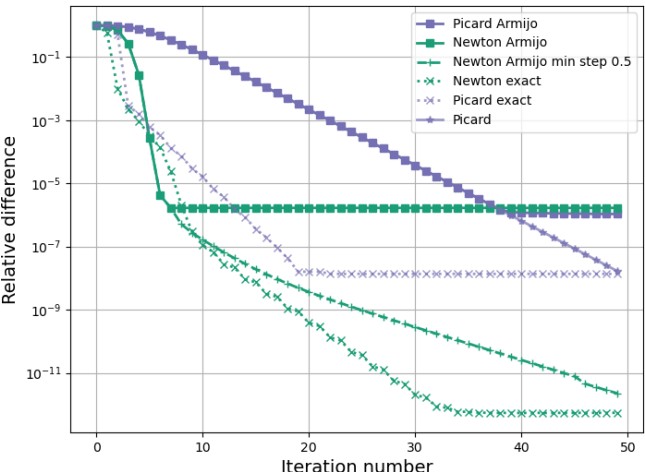

**Figure 4.** Relative difference compared to the reference solution for ISMIP-HOM B.

not so important (Joughin et al., 2010, Sect. 2.3). We use two error measurements because one method could be better for one purpose and the other for another. The local relative difference reflects that regions with small velocities should also be represented with a small relative error. Both error mea-

surements consider the velocity field for the whole domain of the glacier. In contrast, the original experiment (Pattyn et al., 2008) only considers the velocity field at the surface.

Figure 4 displays the relative difference over the iteration number. First, we discuss the Picard variants. Without a step size control, the convergence rate is slow. The method needs 39 iterations to obtain a reduction to $10^{-6}$. Exact step sizes lead to a faster reduction in the relative difference.

For Newton's method, the exact step sizes reduce the relative difference really fast. We see this even better if we consider just the first nine iterations (Fig. 5). The Armijo step sizes obtain the reduction in the Picard iteration without a step size control after only seven iterations. This reduces the necessary number of iterations by 82 %.

However, after a few iterations, the relative difference does not reduce anymore. This could be seen as either a mistake in the step size control or Newton's method. However, the Picard iteration with Armijo step sizes has the same problem. It is identical to the Picard iteration without a step size control up to iteration 39 and then chooses smaller step sizes to stall at a similar difference as Newton's method. Imposing a minimum step size of 0.5 helps to circumvent this problem. Newton's method then reduces the relative difference up to iteration 39.

We think the reason for this behavior is that the discretized minimum of the convex function, $J$, and the root of the full-Stokes equations are slightly different. To verify this claim, we solve the full-Stokes equations on one grid and refine-

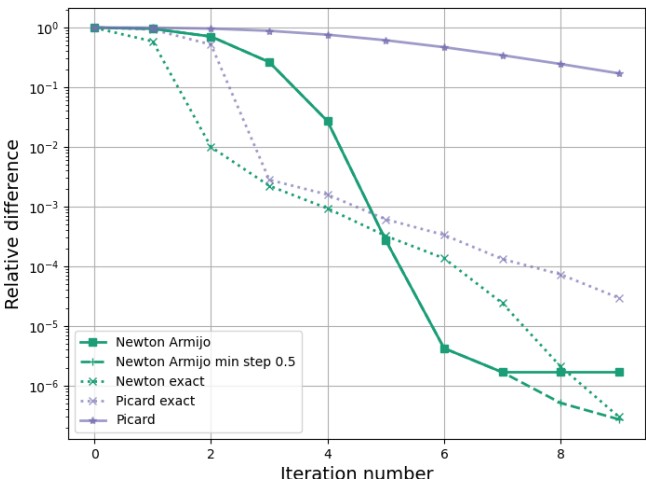

**Figure 5.** Relative difference compared to the reference solution for ISMIP-HOM B for the first 9 iterations.

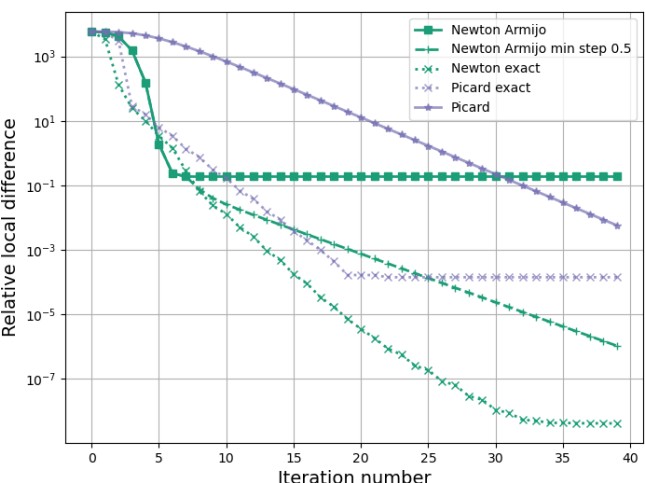

**Figure 7.** Relative local difference compared to the reference solution for ISMIP-HOM B.

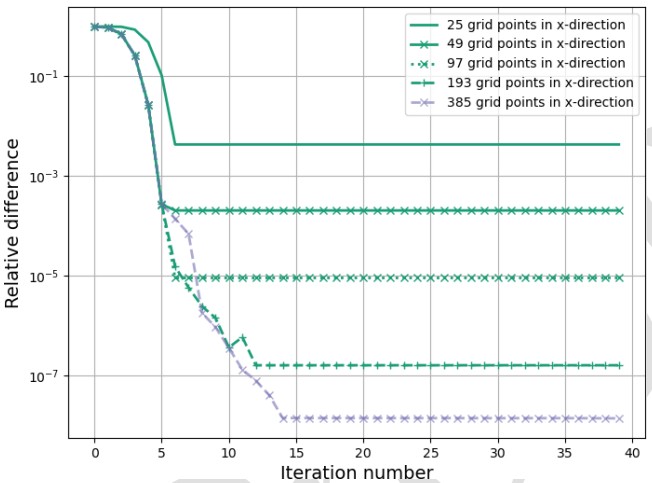

**Figure 6.** Experiments with 25, 49, 97, 193, and 385 grid points in $x$ direction and 3, 5, 9, 17, and 34 grid points in $z$ direction for the experiment ISMIP-HOM B for Newton's method with Armijo step sizes.

ments of it, ensuring that all grid points on the coarser grids are also on the finer grids. We see the dependence on the resolution in Fig. 6. By halving the grid size, we reduce the relative difference by a bit more than a factor of 10. The increase in the relative difference from iteration 10 to 11 for the resolution with 193 grid points in the $x$ direction seems nonintuitive. But, we remind that the Armijo step size control tries to minimize the function $J$ not find the root.

Also, Hirn (2013) reported accuracy problems for a small value of $\delta$. Hirn considered a channel flow with $\rho \boldsymbol{g} = \boldsymbol{0}$ and $n \in \{2, 10/3, 5, 10\}$. The stopping criterion is not reached for $n = 10$ and for $n = 5$ for higher resolutions. In the second experiment, Hirn introduced $\delta > 0$, with $\delta = \delta_0 h^{2/(1+1/n)}$ and $\delta_0 \in \{1, 10\}$ and the mesh size $h$. Both variants converged to the wanted accuracy. Additionally, the calculated solutions

for all resolutions were not too different from the analytical solution compared to the original problem with $\delta_0 = 0$. Finally, Hirn (2013) counted the number of Newton iterations to reach the wanted accuracy: the variants with $\delta_0 \in \{1, 10\}$ always converge and need a lower number of iterations compared to $\delta_0 = 0$.

Thus, a higher $\delta$ value could lead to the expected quadratic convergence. In contrast, the exact step sizes do not seem to have this problem as they do not rely on evaluating the function $J$. Newton's method with exact step sizes has the advantage that the error reduces even more without using a minimal step size. Thus, one less parameter needs to be selected. Even the Picard iteration with exact step sizes is much better than the Picard iteration without a step size control. It only needs 15 iterations to obtain the accuracy, for which the Picard iteration without a step size control needs 39 iterations. That corresponds to a reduction of 62 %. The latter approach also has the advantage that there is no need to implement a new method to solve the problem. Only the relatively simple calculation of the step sizes needs to be implemented.

The results are really similar for our second measure of the accuracy, the relative local error (see Fig. 7).

All our algorithms are better than the Picard iteration without a step size control in this experiment. The reduction with Newton's approach with both step size controls is 77 % now. The fast convergence is again impressive, especially for the first nine iterations (see Fig. 8).

### 5.4 The experiment ISMIP-HOM A

Because real-world applications are three-dimensional, we consider experiment ISMIP-HOM A. This experiment extends ISMIP-HOM B to three dimensions. All chosen constants are the same as in the experiment ISMIP-HOM B. The experiment ISMIP-HOM A has a sinusoidal bottom in both horizontal dimensions. Again, we have three copies of the

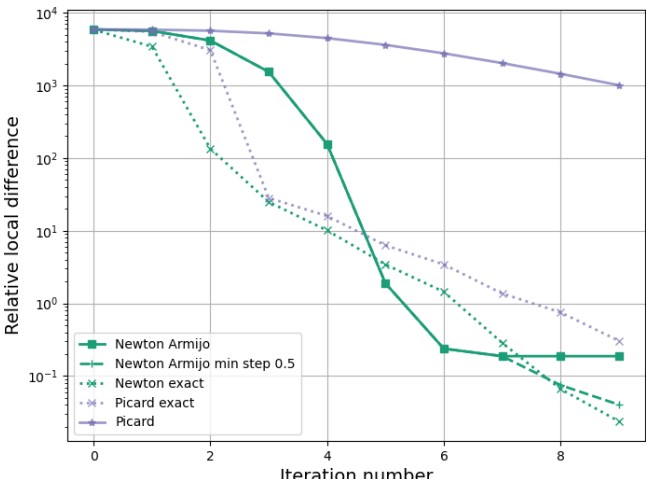

**Figure 8.** Relative local difference compared to the reference solution for ISMIP-HOM B for the first nine iterations.

glacier in both horizontal directions. Thus, we have in total 48 copies. We describe the surface and bottom by

$$z_s(x, y) = -x \tan(\beta),$$
$$z_b(x, y) = z_s(x, y) - 1000 + 500 \cdot \sin(\omega x) \sin(\omega y). \quad (14)$$

### 5.5 Results for experiment ISMIP-HOM A

All our methods produce very similar results and overlap (see Figs. 9 and 10). Our simulations reproduce the surface velocity at $y = L/4$ from Pattyn et al. (2008) for the full-Stokes simulations for the majority of the glacier, but they produce higher velocity values than the mean added to the standard deviation around $x = L/3$. Nonetheless, the maximum relative difference is less than 0.02 (see Fig. 9).

The general convergence behavior for the three-dimensional experiment is similar to the two-dimensional experiment.

However, Newton's method is even better in three dimensions (see Fig. 11). Again, zooming to the first few iterations states the benefit from Newton's method and the step size control more impressing (see Fig. 12). The Picard iteration without a step size control needs 39 iterations to have the same accuracy as Newton's method using Armijo step sizes after 6 iterations. Thus, the necessary number of iterations is reduced by more than 85 %. Again, a minimum step size of $\alpha = 0.5$ helps to reduce the relative difference after a few iterations. The exact step sizes are even better. They decrease the relative difference (see Fig. 11) and the relative local difference (see Fig. 13) further than the Armijo step sizes.

For the Picard iteration, we again see a slow convergence without a step size control, especially if we consider the first few iterations (see Fig. 14). This figure emphasizes that the Picard iteration without a step size control converges slowly compared to the other methods. Exact steps lead to a great improvement.

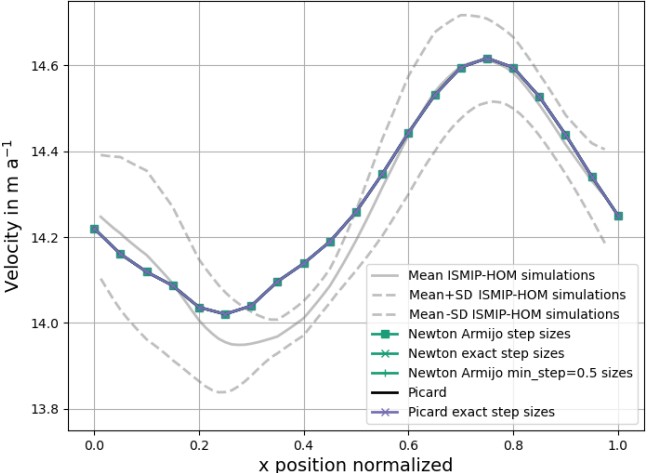

**Figure 9.** Simulated surface velocity at $y = L/4$ in meters per year for different solvers for ISMIP-HOM A. All our calculated velocity fields overlap with each other. The mean and the standard deviation from Pattyn et al. (2008) with five models are plotted in grey. The mean and standard deviations have no values at $x = 0$ and $x = 1$ due to missing values.

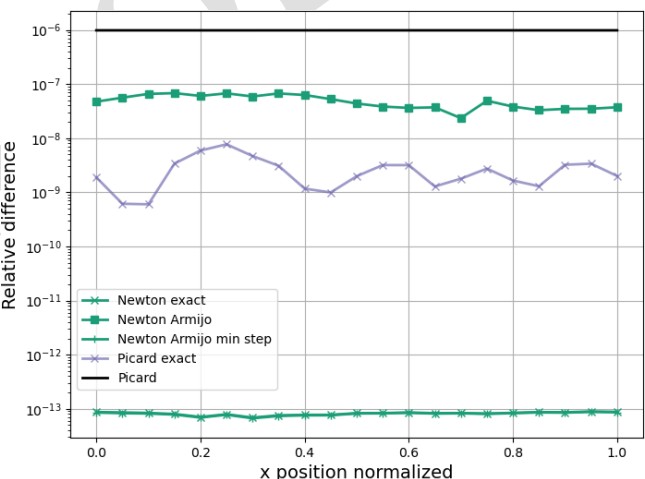

**Figure 10.** Relative difference in $|v - v_{ref}|/|v_{ref}|$ for each grid point at the surface at $y = L/4$ for ISMIP-HOM A. The reference solution is the solution from 80 Picard iterations without a step size control. The relative difference for Newton with exact step sizes and Newton with Armijo step sizes and a minimal step of 0.5 are nearly identical.

As this experiment is more realistic regarding the number of grid points, we calculated the computation time for each iteration in experiment ISMIP-HOM A (see Table 1). There are two key findings: The computation times for the Newton variants are about 20 % higher than for the Picard variants. Additionally, the step size control is computationally cheap compared to the rest of Newton's method or the Picard iteration.

However, the three-dimensional experiment has the additional uncertainty in precise computation times as we used

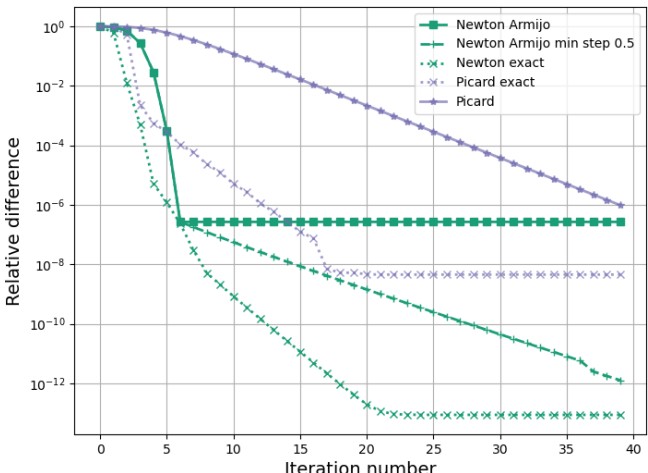

**Figure 11.** Relative difference compared to the reference solution for ISMIP-HOM A.

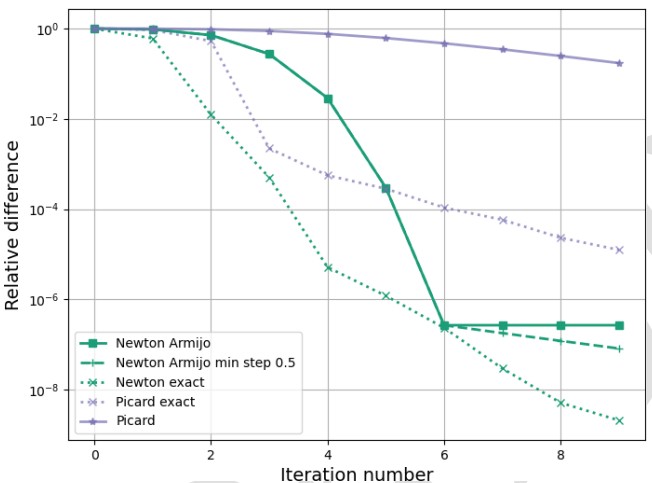

**Figure 12.** Relative difference compared to the reference solution for ISMIP-HOM A for the first nine iterations.

the same processor type but a different processor on the high-performance computer.

## 6 Instationary experiments

In this section, we simulate a time-dependent version of the Haut Glacier d'Arolla without and with sliding. In a first step, we verify that our model produces similar results as in the experiments ISMIP-HOM E1 and E2 (see Pattyn et al., 2008). The top and the bottom of the glacier are given by an input file. At the bottom, we have Dirichlet boundary conditions, and at the top, $\boldsymbol{\sigma} \cdot \boldsymbol{n} = \boldsymbol{0}$. The domain is represented in Fig. 15. In contrast to the stationary problems in Sect. 5, we do not have a reference solution. A stopping criterion in Gagliardini and Zwinger (2008, Sect. 2.4.2) is

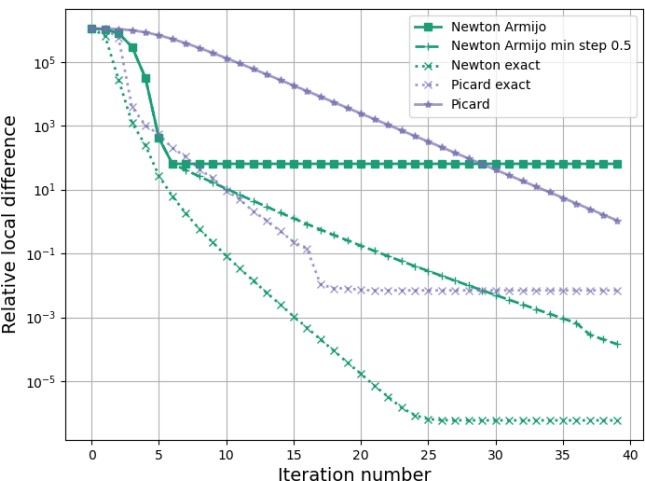

**Figure 13.** Relative local difference compared to the reference solution for ISMIP-HOM A.

$$2\frac{|\boldsymbol{v}_{k+1} - \boldsymbol{v}_k|}{|\boldsymbol{v}_{k+1}| + |\boldsymbol{v}_k|} < 10^{-5}. \tag{15}$$

To include the step size, we would multiply the right-hand side with $\alpha$. This criterion is suitable for stopping the iteration in a real simulation if the velocity field only has small changes. However, checking the relative difference let the Newton variants with step size control stop earlier in our test simulation without decreasing the error $\|G(\boldsymbol{v}, p)\|$ as much as the Picard variants did. As we compare solvers, we need to check if $G(\boldsymbol{v}_k, p_k)$ is close enough to zero. Thus, we check if we are close enough to our solution compared to the initial guess:

$$\|G(\boldsymbol{v}_k, p_k)\|/\|G(\boldsymbol{v}_0, p_0)\| < \epsilon,$$

with $\epsilon := 10^{-3}$. A relative stopping criterion seems necessary to reduce dependence in the domain and the absolute velocities. Thus, the calculated velocity field should have an error of 0.1 % compared to the initial guess after each time step. Our initial guess for time-dependent problems is the solution of a Stokes problem before the first step. After calculating a velocity field, the surface velocity determines the new surface. The grid points are moved according to the new surface. The initial guess for the velocity on the new domain is our velocity field shifted to the new domain. The stopping criterion has the advantage that we count the number of iterations needed to reduce the error by a certain factor. Therefore, the wanted error reduction is the same for all our solvers.

We know $G(\boldsymbol{v}_k, p_k) \in (H \times L)^*$. The Riesz isomorphism yields the existence of $(\tilde{\boldsymbol{v}}_k, \tilde{p}_k) \in H \times L$ with

$$\int_{\Omega} \nabla \tilde{\boldsymbol{v}}_k : \nabla \boldsymbol{\phi} \, \mathrm{d}x + \int_{\Omega} \mathrm{div}(\boldsymbol{\phi}) \tilde{p}_k \, \mathrm{d}x + \int_{\Omega} q \, \mathrm{div}(\tilde{\boldsymbol{v}}_k) \, \mathrm{d}x =$$

$$\langle G(\boldsymbol{v}_k, p_k), (\boldsymbol{\phi}, q) \rangle_{V_2^*, V_2} \quad \text{for all } (\boldsymbol{\phi}, q) \in H \times L. \tag{16}$$

**Table 1.** Computation time in seconds without diagnostic calculations like the residual norm for the complete iteration.

| | Complete iteration | | Step size calculation | |
|---|---|---|---|---|
| | Mean | Standard deviation | Mean | Standard deviation |
| Picard without a step size control | 2226 | 20.3 | – | – |
| Picard with exact step sizes | 2286 | 11.0 | 60.4 | 0.18 |
| Newton with Armijo step sizes and minimum step size of 0.5 | 2706 | 6.77 | 4.60 | 0.45 |
| Newton with exact step sizes | 2757 | 18.5 | 60.1 | 0.24 |

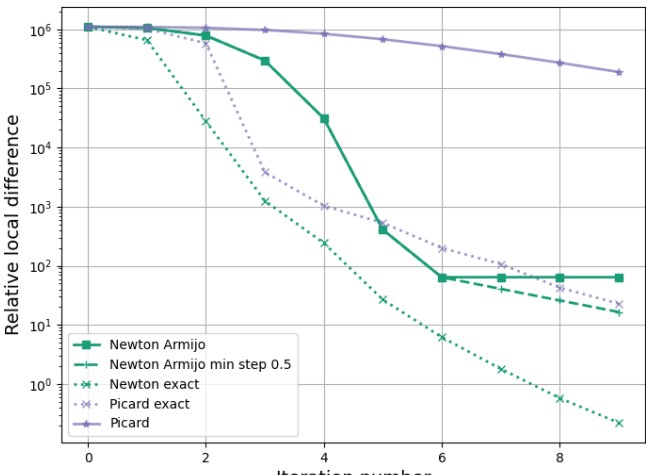

**Figure 14.** Relative local difference compared to the reference solution for ISMIP-HOM A for the first nine iterations.

Thus, we have to solve another Stokes problem in each iteration. Note that this Stokes problem is only diagnostic, and we do not need to solve it in practice. As the numerical analysis focuses on the velocity field, we calculate our error by

$$\|\tilde{\boldsymbol{v}}_k\|_{V_2} = \sqrt{\int_\Omega |\nabla \tilde{\boldsymbol{v}}_k|^2 \, dx}. \tag{17}$$

For the experiment with sliding, we have to handle a difficulty arising in FEniCS: we can only force the boundary condition $\boldsymbol{v} \cdot \boldsymbol{n} = 0$ on horizontal and vertical boundaries. Thus, we use nine small stairs at the bottom instead of the slope in the original problem (see Fig. 16). On the bottom boundary with $2200 < x < 2500$, we employ the boundary condition $v_z = 0$ for $\boldsymbol{v} = (v_x, v_z)$.

## 6.1 Stationary solutions

We only discuss the accuracy of our model in simulating the experiments ISMIP-HOM E1 and E2 without considering convergence speed. We discuss the convergence speed for these time-dependent problems. The simulation of the velocity field is quite similar to the results in Pattyn et al. (2008) (see Fig. 17). Our velocity field at the surface is mostly within

the mean with the standard deviation of the reference solutions (see Fig. 17). In some small parts, the velocity is slightly lower.

For the experiment with sliding, the calculated velocity field is near the mean minus the standard deviation of the reference solutions in Pattyn et al. (2008) (see Fig. 18).

Often, it is even a bit less. However, it is still a suitable approximation, and we use both problems for the time-dependent simulation.

## 6.2 Time-dependent problem – mass transport

For the instationary problem, the surface develops dependent on the velocity field. In our case, we describe the height of the glacier by Eq. (18) (see Pattyn et al., 2008).

$$\frac{\partial z(x)}{\partial t} + v_x(x)\frac{\partial z(x)}{\partial x} - v_z(x) = 0 \quad \text{for } x \in (0, 5000] \tag{18}$$

The height is fixed at $x = 0$. Let $(x_i)_{i=0}^N$ be the discretization with $x_0 = 0$ and $x_N = 5000$. We approximate the spatial differential quotient by an upwinding scheme:

$$v_x(x_i)\frac{\partial z(x_i)}{\partial x} \approx$$
$$\begin{cases} v_x(x_i)\frac{z(x_i)-z(x_{i-1})}{x_i-x_{i-1}} & \text{for } v_x(x_i) > 0 \text{ and } i > 0, \\ v_x(x_i)\frac{z(x_{i+1})-z(x_i)}{x_{i+1}-x_i} & \text{for } v_x(x_i) \le 0 \text{ and } i < N. \end{cases} \tag{19}$$

The upwinding scheme stabilizes the solution of the discretized Eq. (18). Moreover, it helps in our experiments for the conversation of mass compared to the forward difference quotient and the central difference quotient. We use an explicit Euler method in time and conclude for the $(k+1)$th time step:

$$\frac{z^{k+1} - z^k}{\Delta t} + v_x^k(x)\frac{\partial z^k(x)}{\partial x} - v_z^k(x) = 0. \tag{20}$$

Together, we obtain the following for the $(k+1)$th time step and the $i$th grid point at the surface:

$$z_i^{k+1} = z_i^k + \Delta t \left( v_z^k(x_i) - \right.$$
$$\left. \begin{cases} v_x^k(x_i)\frac{z^k(x_i)-z^k(x_{i-1})}{x_i-x_{i-1}} & \text{for } v_x(x_i) > 0 \text{ and } i > 0, \\ v_x^k(x_i)\frac{z^k(x_{i+1})-z^k(x_i)}{x_{i+1}-x_i} & \text{for } v_x(x_i) \le 0 \text{ and } i < N, \\ 0 & \text{for } i = 0. \end{cases} \right) \tag{21}$$

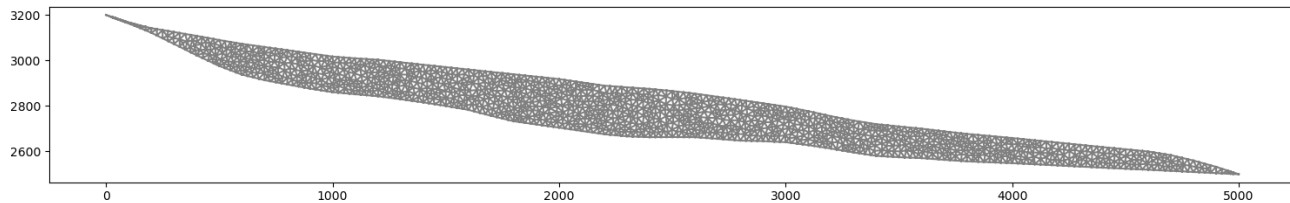

**Figure 15.** Domain of the Haut Glacier d'Arolla.

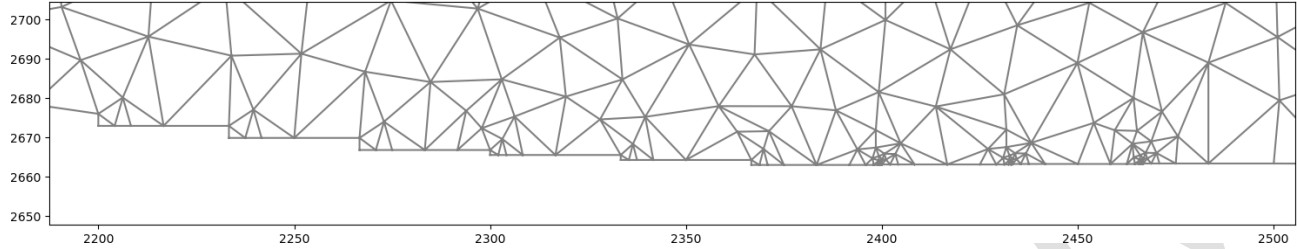

**Figure 16.** Stair-shaped domain at the bottom.

In our problem, the value $z(x_0)$ is fixed. Mathematically, we are not allowed to fix $z(x_N)$ because this value is determined by Eq. (18). Therefore, we add a grid point at (5000, 2505), slightly above the bottom at (5000, 2500). We impose $\boldsymbol{\sigma} \cdot \boldsymbol{n} = \boldsymbol{0}$ on the newly generated right boundary. Hence, the mass can flow outside the glacier or physically interpreted ice is melting.

We calculate over 30 years to simulate a changing velocity field with the highest surface velocity at the right edge of the domain. We choose a time step size of 0.25 years to fulfill the CFL condition for experiment E1. In experiment E2, we choose the same time step sizes to have a comparable experiment.

## 6.3 Time-dependent simulation without friction

In this subsection, we visualize the velocity field of the glacier at the surface over the time simulation and discuss the computational effort for the experiment without friction.

All simulations produce similar surface velocities over time (see Fig. 19).

Thus, all methods seem to calculate the solution appropriately. Now, we discuss the computational effort. The number of iterations needed is shown in Fig. 20. The corresponding mean and standard deviations are in Table 2.

First, we discuss Newton's method. Armijo step sizes with a minimum step size of 0.5 have the problem that they do not always converge. This also leads to the largest standard deviation. The exact step sizes always converge and have the lowest mean. The standard deviation is still larger than for the Picard variants.

Second, we discuss the Picard variants. Without a step size control, the mean number of iterations is the highest of all four algorithms. The standard deviation is the lowest, and the

**Table 2.** Number of iterations for solving the full-Stokes equations on each time step.

| | Time steps | Mean | Standard deviation |
|---|---|---|---|
| Picard without a step size control | 119 | 16.79 | 0.41 |
| Picard with exact step sizes | 119 | 9.29 | 0.46 |
| Newton with Armijo step sizes and minimum step size of 0.5 | 119 | 10.92 | 10.67 |
| Newton with exact step sizes | 119 | 6.45 | 2.83 |

number of iterations only fluctuates by one. The exact step sizes reduce the necessary number of iterations by more than 40 % and only need about two iterations more than Newton's method with these step sizes. The standard deviation is nearly the same.

We measured the computation time for the time-dependent experiment (see Table 3).

The computation of the step sizes takes 9 % or less of the total computation time for each iteration. The computation time for calculating diagnostics like the residual norm was not measured as it is unnecessary for the application.

## 6.4 Time-dependent simulation with friction

In this subsection, we visualize the velocity field of the glacier at the surface over the time simulation and discuss the computational effort for the experiment with friction.

All simulations produce similar surface velocities over time (see Fig. 21). Thus, all methods seem to calculate the solution appropriately. Now, we discuss the computational effort. The number of iterations needed is shown in Fig. 22.

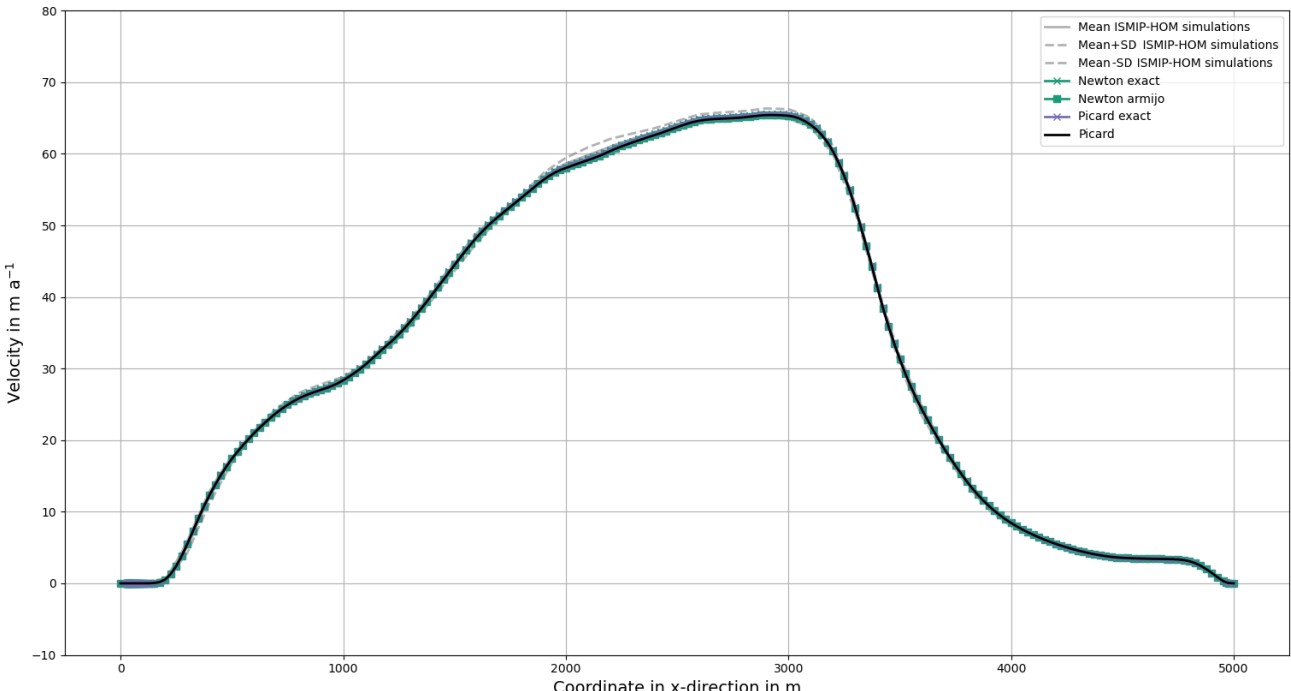

**Figure 17.** Surface velocity field of the Haut Glacier d'Arolla without sliding.

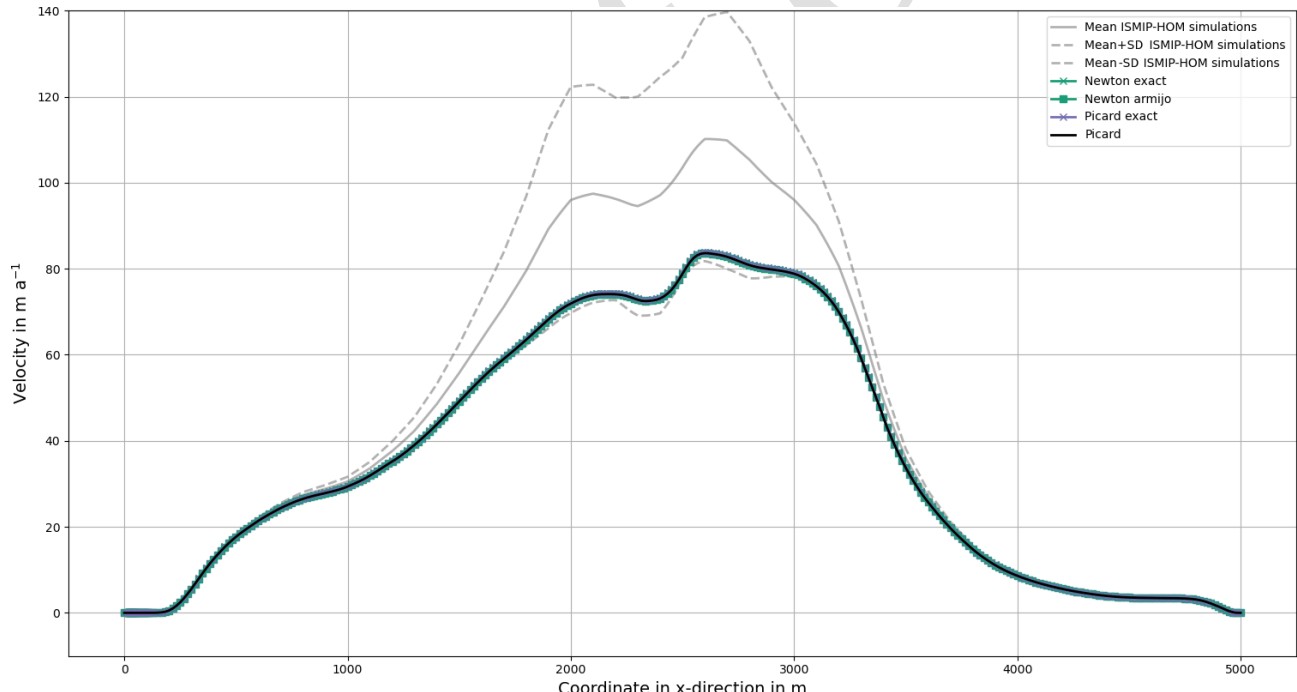

**Figure 18.** Surface velocity field of the Haut Glacier d'Arolla with sliding.

The corresponding mean and standard deviations are in Table 4.

First, we discuss Newton's method. Armijo step sizes with a minimum step size of 0.5 have the problem that they often do not converge. This also leads to the largest mean number of iterations and the highest standard deviation. The exact step sizes do not converge in two cases. This also leads to a large standard deviation.

Second, we discuss the Picard variants. Without a step size control, the necessary number of iterations is nearly always

**Table 3.** Computation time in seconds for the complete iteration without diagnostic calculations like the residual norm.

| | Iterations | Complete iteration | | Step size calculation | |
|---|---|---|---|---|---|
| | | Mean | Standard deviation | Mean | Standard deviation |
| Picard without a step size control | 1998 | 5.61 | 0.61 | – | – |
| Picard with exact step sizes | 1106 | 5.76 | 0.58 | 0.22 | 0.01 |
| Newton with Armijo step sizes and minimum step size of 0.5 | 1299 | 5.73 | 0.60 | 0.02 | 0.01 |
| Newton with exact step sizes | 767 | 6.01 | 0.57 | 0.48 | 0.03 |

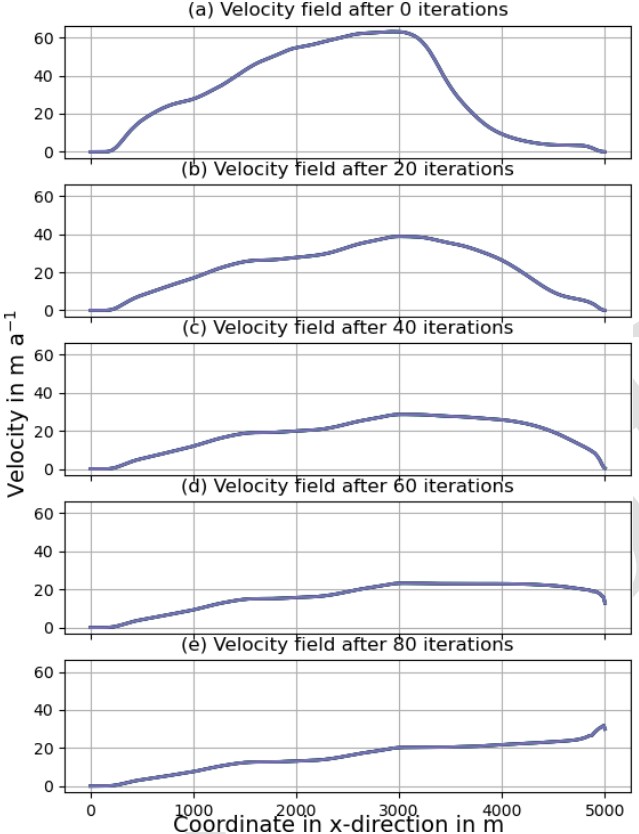

**Figure 19.** Surface velocity fields for the Picard iteration without a step size control, the Picard iteration with exact step sizes, Newton's method with Armijo step sizes, and Newton's method with exact step sizes. The velocity fields for the different methods are nearly identical.

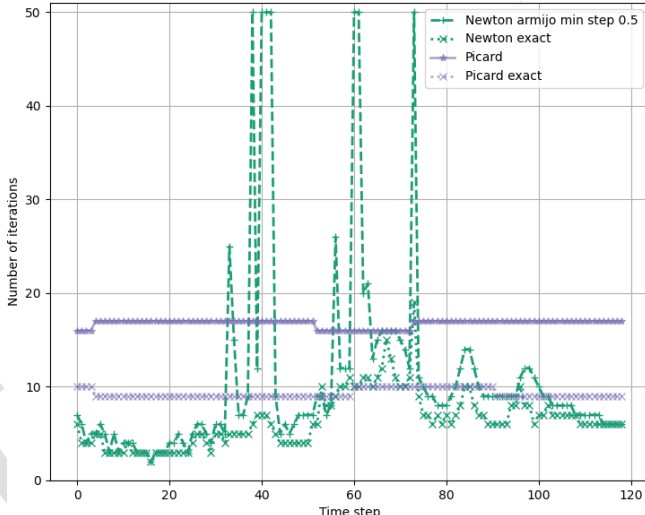

**Figure 20.** The number of iterations needed to solve the full-Stokes equations between each time step. The maximum number of iterations in each time step is set to 50.

**Table 4.** Number of iterations for solving the full-Stokes equations in each time step.

| | Time steps | Mean | Standard deviation |
|---|---|---|---|
| Picard without a step size control | 119 | 16.72 | 0.55 |
| Picard with exact step sizes | 119 | 9.39 | 0.49 |
| Newton with Armijo step sizes and minimum step size of 0.5 | 119 | 24.70 | 20.64 |
| Newton with exact step sizes | 119 | 10.42 | 7.48 |

the same but relatively large. The use of exact step sizes leads to the smallest mean and standard deviations. Thus, this method is the most efficient and, with a small standard deviation, also reliable. The exact step sizes for the Picard iteration are still better than for Newton's method without the two time steps that do not converge. The Picard variants also need nearly the same number of iterations for the experiment with and without sliding in contrast to the Newton variants.

We measured the computation time for the time-dependent simulation (see Table 5). The computation time for calculating diagnostics like the residual norm was not measured as it is unnecessary for the application.

The computation of the step sizes takes 6.5 % or less of the total computation time for each iteration.

**Table 5.** Computation time in seconds for the complete iteration without diagnostic calculations like the residual norm.

|  | Iterations | Computation time each iteration | | Computation time step size | |
|---|---|---|---|---|---|
|  |  | Mean | Standard deviation | Mean | Standard deviation |
| Picard without a step size control | 1990 | 10.93 | 0.77 | – | – |
| Picard with exact step sizes | 1118 | 11.31 | 0.85 | 0.30 | 0.01 |
| Newton with Armijo step sizes and minimum step size of 0.5 | 2939 | 11.22 | 0.86 | 0.03 | 0.01 |
| Newton with exact step sizes | 1240 | 11.62 | 0.75 | 0.67 | 0.03 |

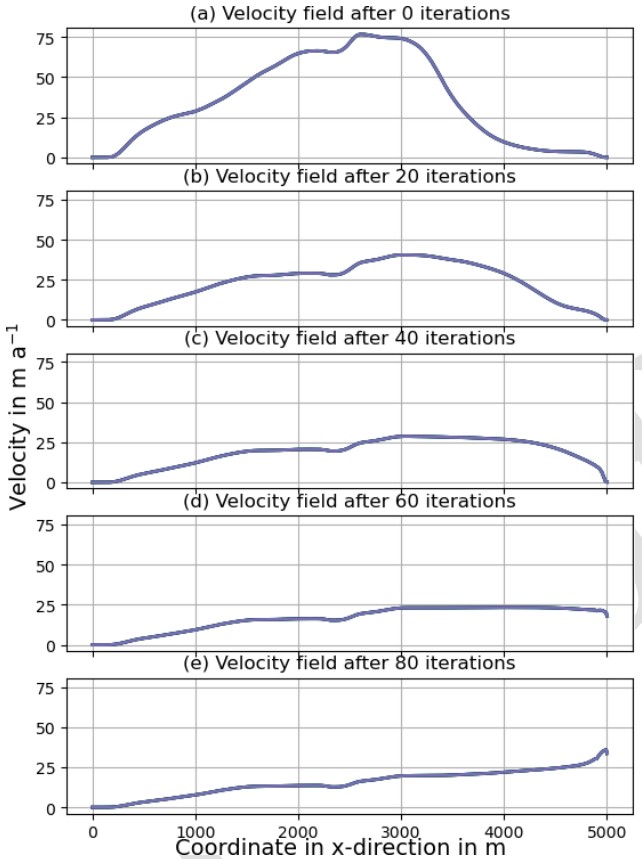

**Figure 21.** Surface velocity fields for the Picard iteration without a step size control, the Picard iteration with exact step sizes, Newton's method with Armijo step sizes, and Newton's method with exact step sizes. The velocity fields for the different methods are nearly identical.

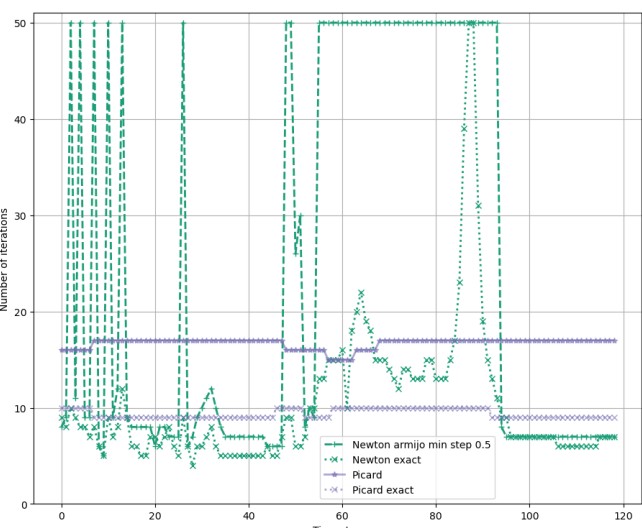

**Figure 22.** Number of iterations to solve the full-Stokes equations between each time step. The maximum number of iterations in each time step is set to 50.

dependent variant of the Glacier d'Arolla experiment with and without sliding.

We observe that our calculated solutions are similar to those in Pattyn et al. (2008). However, the approximately exact step sizes greatly improve the convergence speed of the Picard iteration and ensure the convergence of Newton's method for nearly all situations except two cases in the time-dependent simulation with friction (see Fig. 22). Thus, the approximately exact step sizes seem to be better than the Armijo step sizes.

The computation time of the step sizes is only a small part of the complete iteration. In our experiments, the time for calculating the step sizes took 9 %, 6.5 %, and 3 %. The ratio is even smaller for higher resolutions. The concrete computation times in seconds are irrelevant as they depend on the hardware.

## 7 Summary and conclusion

Solving the full-Stokes equations is equivalent to minimizing a function. We use this function to introduce approximately exact step sizes. For a comparison, we also use Armijo step sizes. We test the algorithms for benchmark experiments with a sinusoidal bottom in two and three dimensions, for the Glacier d'Arolla benchmark experiment, and for a time-

## 8 Outlook

The effort to implement the algorithms above is relatively low. For every boundary condition added to those above, one has to check if a convex function exists. One only needs to implement these convex functions, the directional derivatives, and the Armijo and exact step sizes, respectively. The Picard iteration without a step size control or Newton's method should already be implemented for solving the full-Stokes equations.

There are a few possible directions to work on: the computation of the step size could be done more efficiently by parallelizing the calculation of the integrals and testing how many bisections are necessary for calculating the exact step sizes.

Also, more realistic three-dimensional examples or different sliding laws could be tested. The mathematical theory for a nonlinear sliding boundary condition is discussed in Schmidt (2023).

The implementation in ice models is another way to check if the presented algorithms work in real-world applications. Lastly, the step size control might reduce the number of iterations for the higher-order equations. Solving those equations is also equivalent to finding the minimum of a convex function (see Schoof, 2010).

## Appendix A: Mathematical derivations

### A1 The variational formulation

For the well-posedness of the Picard iteration without a step size control (see Algorithm 1), the following applies:

$$B\left(|D\boldsymbol{v}_k|^2 + \delta^2\right)^{\frac{1-n}{2n}} D\boldsymbol{v}_{k+1} : \nabla\boldsymbol{\phi} \tag{A1}$$

has to be integrable. Due to a bounded ice rheology $B$ and $\delta > 0$, follows the boundedness with $c \in \mathbb{R}$ and

$$B\left(|D\boldsymbol{v}_k|^2 + \delta^2\right)^{\frac{1-n}{2n}} < c. \tag{A2}$$

Thus, we need, for integrability, $D\boldsymbol{v}_{k+1} : \nabla\boldsymbol{\phi} \in L^1(\Omega)$. This is only fulfilled for $D\boldsymbol{v}_{k+1} \in L^p(\Omega)^{N\times N}$ and $\nabla\boldsymbol{\phi} \in L^q(\Omega)^{N\times N}$ with $1/p + 1/q = 1$. Hence, we can not use $\boldsymbol{v}_{k+1}, \boldsymbol{\phi} \in \{\boldsymbol{v} \in W^{1,1+1/n}(\Omega)^N; \boldsymbol{v}|_{\Gamma_b \cup \Gamma_\ell} = \boldsymbol{0}\}$ (see Belenki et al., 2012, Sect. 2.3), which is the suitable space for $\mu_0 = 0$ as it allows for the proof of existence and uniqueness of the solution.

However, expression (A1) is well defined for $\boldsymbol{v}_{k+1}, \boldsymbol{\phi} \in H = \{\boldsymbol{v} \in H^1(\Omega)^N; \boldsymbol{v}|_{\Gamma_b \cup \Gamma_\ell} = \boldsymbol{0}\}$. The additional diffusion term with $\mu_0 > 0$ verifies that the solution of the full-Stokes equations is in $H$. Similar reasons make the diffusion term necessary for Newton's method; the directional derivative (see Eq. 6) is only defined for $\boldsymbol{v}, \boldsymbol{w}, \boldsymbol{\phi} \in H$.

### A2 The directional derivative of $G$

In this subsection, we compute the derivative of $G$ at the velocity $\boldsymbol{v} \in H$ and pressure $p \in L$ in the direction $\boldsymbol{w} \in H$ and $q \in L$, with the diffusion $\mu_0 > 0$. Because we have a variational formulation, we can only interpret this derivative for test functions $\boldsymbol{\phi} \in H$ and $\psi \in L$. We calculate

$$\langle G'(\boldsymbol{v}, p)(\boldsymbol{w}, q), (\boldsymbol{\phi}, \psi)\rangle$$
$$= \lim_{t \to 0} \frac{\langle G(\boldsymbol{v} + t\boldsymbol{w}, p + tq), (\boldsymbol{\phi}, \psi)\rangle - \langle G(\boldsymbol{v}, p), (\boldsymbol{\phi}, \psi)\rangle}{t}$$
$$= \lim_{t \to 0} \int_\Omega \frac{B}{t}\left(\left(|D(\boldsymbol{v} + t\boldsymbol{w})|^2 + \delta^2\right)^{\frac{1-n}{2n}} D\boldsymbol{v}\right.$$
$$\left. - \left(|D\boldsymbol{v}|^2 + \delta^2\right)^{\frac{1-n}{2n}} D\boldsymbol{v}\right) : \nabla\boldsymbol{\phi}\,\mathrm{d}x$$
$$+ \lim_{t \to 0} \int_\Omega \frac{B}{t}\left(\left(|D(\boldsymbol{v} + t\boldsymbol{w})|^2 + \delta^2\right)^{(1-n)/(2n)} t D\boldsymbol{w}\right) : \nabla\boldsymbol{\phi}\,\mathrm{d}x$$
$$- \lim_{t \to 0}\left(\int_\Omega \frac{p + tq - p}{t}\mathrm{div}\boldsymbol{\phi}\,\mathrm{d}x + \int_\Omega \mathrm{div}\left(\frac{\boldsymbol{v} + t\boldsymbol{w} - \boldsymbol{v}}{t}\right)\psi\right)\mathrm{d}x. \tag{A3}$$

The limits for the last three lines on the right-hand side of the last equality are clear. For the first line, we use the Taylor expansion. Therefore, we define the function $f_x : [0, \infty) \to \mathbb{R}$ as

$$f_x(t) = \left(|D(\boldsymbol{v}(x) + t\boldsymbol{w}(x))|^2 + \delta^2\right)^{\frac{1-n}{2n}}. \tag{A4}$$

Its derivative is

$$f_x'(t) = \frac{1-n}{n}\left(|D(\boldsymbol{v}(x) + t\boldsymbol{w}(x))|^2 + \delta^2\right)^{\frac{1-3n}{2n}}$$
$$\left(D\boldsymbol{v}(x) : D\boldsymbol{w}(x) + t|D\boldsymbol{w}(x)|^2\right). \tag{A5}$$

We calculate the derivative by assuming we can draw the limes into the integral. A detailed explanation of why we can do this is in Schmidt (2023). We obtain the following with $\xi : \Omega \to [0, t]$ for the Taylor expansion:

$$\int_\Omega \lim_{t \to 0} \frac{B}{t}\left(\left(|D(\boldsymbol{v} + t\boldsymbol{w})|^2 + \delta^2\right)^{\frac{1-n}{2n}} D\boldsymbol{v}\right.$$
$$\left. - \left(|D\boldsymbol{v}|^2 + \delta^2\right)^{\frac{1-n}{2n}} D\boldsymbol{v}\right) : \nabla\boldsymbol{\phi}\,\mathrm{d}x$$
$$= \int_\Omega \lim_{t \to 0} \frac{B}{t}\left(f_x(t) - f_x(0)\right) D\boldsymbol{v} : \nabla\boldsymbol{\phi}\,\mathrm{d}x$$
$$= \int_\Omega \lim_{t \to 0} \frac{B}{t} f_x'(\xi(\boldsymbol{x})) t D\boldsymbol{v} : \nabla\boldsymbol{\phi}\,\mathrm{d}x$$
$$= \int_\Omega \lim_{t \to 0} B\frac{1-n}{n}\left(|D\boldsymbol{v}(x) + \xi(x)\boldsymbol{w}(x)|^2 + \delta^2\right)^{\frac{1-3n}{2n}}$$
$$\left(D\boldsymbol{v}(\boldsymbol{x}) : D\boldsymbol{w}(\boldsymbol{x}) + \xi(\boldsymbol{x})|D\boldsymbol{w}(\boldsymbol{x})|^2\right) D\boldsymbol{v}(\boldsymbol{x}) : \nabla\boldsymbol{\phi}(\boldsymbol{x})\,\mathrm{d}x$$
$$= \int_\Omega B\frac{1-n}{n}\left(|D\boldsymbol{v}|^2 + \delta^2\right)^{\frac{1-3n}{2n}}(D\boldsymbol{v} : D\boldsymbol{w})(D\boldsymbol{v} : \nabla\boldsymbol{\phi})\,\mathrm{d}x. \tag{A6}$$

*Code and data availability.* The model is available at https://doi.org/10.5281/zenodo.10979366 (Schmidt, 2024).

The latest version of the source code is available at https://github.com/Niko-ich/FEniCS-full-Stokes (last access: 13 June 2024).

*Author contributions.* The paper is written by NS with contributions and suggestions from TS and AH. The code is implemented by NS with support from TS and AH.

*Competing interests.* The contact author has declared that none of the authors has any competing interests.

*Acknowledgements.* The authors appreciate helpful explanations of the ISMIP-HOM experiments from Martin Rückamp from the Bavarian Academy of Sciences and Humanities and Thomas Kleiner from Alfred-Wegener-Institut in Bremerhaven. We also thank Mathieu Morlighem from Dartmouth College in Hanover, New Hampshire, for answering questions about ISSM and Fabien Gillet-Chaulet from the Université Grenoble Alpes for answering questions about Elmer/Ice. We thank the reviewers for suggesting experiments to increase the relevance of the paper and other helpful ideas to improve the paper.

*Financial support.* We acknowledge financial support by DFG within the funding programme Open Access-Publikationskosten.

*Review statement.* This paper was edited by Ludovic Räss and reviewed by Ludovic Räss and one anonymous referee.

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

**Remarks from the typesetter**

TS1  Thank you for the update, please check that the equation is now correct.