# Peer review of "Assessing the benefits of approximately exact step sizes for Picard and Newton solver in simulating ice flow (FEniCS-full-Stokes v.1.3.1)"

_EGUsphere, 2023_

## Author Comment (AC1)

We thank you a lot for the helpful comments and ideas. As suggested, we added the glacier d'Arolla from the ISMIP-HOM experiments in a time-dependent formulation. We also summarized time stepping methods for full-Stokes models. We modified the title to reflect the good performance of the Picard iteration with the approximately exact step sizes.

All upcoming line and page references refer to the new manuscript.

**Summary:** The paper implements a Newton method with step-size control for solving a full Stokes model describing the dynamics of glacial ice. In idealised diagnostic numerical experiments two types of step size control are compared with each other and to Picard iterations with and without step-size control.

**General comments:**

The chosen topic is important and the paper is well structured and seemingly without any obvious technical mistakes. My main concern is about the novelty of this paper, for the following reasons:

1. The Newton method is faster than Picard is expected and Newtons method is already used in ice sheet models extensively, even with step size control (e.g. in Elmer/Ice). It is true that Newton iterations do not converge for all cases especially without running a few initial Picard iterations, but for the numerical experiments chosen in this paper my guess is that Newton works in most existing codes already. Perhaps exact step size is new but I think in that case a brief summary of the most common step size control methods in the most used ice sheet models should be provided. Furthermore, a significant superiority of exact step-sizes compared to Armijo is not clear from this paper.

   We added a summary of the step size controls and methods (Newton/Picard) used in full-Stokes models (lines 23-31). As recommended later, we added time-dependent experiments (pages 14-23).

2. In comparison to the previous manuscript which is referred to in this paper: Schmidt, 2013 https://arxiv.org/pdf/2307.02930.pdf In Schmidt, 2013 more mathematical details and proves are given, but the method and experiments seems identical. Is Schmidt, 2013 indented to be published?

   We intend to publish this manuscript. In fact, a new version with some corrections is in the second review round at *Numerical Algorithms*. A preprint is on *ResearchSquare* at https://www.researchsquare.com/article/rs-3354498/v1 . Due to your feedback, we added time-dependent experiments (pages 14-23) and references to glaciology models (lines 23-31) to this manuscript. These changes make this manuscript more different to the mathematical manuscript.

Nevertheless, I think the paper can be published after major revisions. My suggestions for addressing my above concerns are:

1. If Schmidt, 2013 https://arxiv.org/pdf/2307.02930.pdf is not already submitted somewhere, I think the mathematical proofs here are novel enough for publication. I am not sure if the Cryosphere is the best journal for this, but possibly the mathematical proofs could be explained and summarized in the main text, and the proofs themselves be kept in the appendix.

2. Otherwise, a solution which probably fits the Cryosphere better, is to extend the numerical experiments and literature study so that the importance to glaciology is clearer. If the authors can show that it is possible to use Newtons method in realistic simulations in an effective way without initial Picard iterations, I think it is a major contribution to the community.

We follow this approach.

Specifically the following points should be included:

(a) In the introduction, go through the most important ice sheet models (especially those using the full Stokes model) and review what method they are using to resolve the non-linearity of the problem

We summarized in the introduction, how *ElmerIce*, *ISSM*, *COMSOL multipyhsics*, and *FELIX-S* solve the full-Stokes equations (lines 23-31).

(b) Extend the numerical experiments. If the aim of this paper is to show that some method is better than another and could be useful for an ice sheet modeller, it must be for a relevant example. Most importantly, a time-dependancy (free surface movement) should be introduced. This is important since the first time-step (the only one studied here) usually behaves quite differently in terms of number of iterations, as compared to later time-steps when free surface has relaxed. In a real application, there is also a contact problem between the ice and bedrock, which introduces an extra non-linearity. The more realistic example (3D, variable bedrock) the better. Also, I think CPU-times should be measured rather than only number of iterations.

We added the two-dimensional glacier d'Arolla with variable surface and also with a sliding-part as in the ISMIP-HOM experiments $E1$ and $E2$. We measured the computation times for the time-dependent experiments and the three-dimensional experiment.

If the original paper Schmidt, 2013 https://arxiv.org/pdf/2307.02930.pdf is kept as a separate paper, I think still some of the mathematical details must be added to this paper. I am lacking some mathematical justification of some of the key points of the paper, for instance the justification of the extra diffusion term in the equations, and explanation behind Algorithm 3.

We rearranged the section about step size controls (pages 5-6). We introduced the Armijo step size (lines 122-129) first. Then, we motivate the exact step sizes (lines 130-142). Additionally, we explained the algorithm for the exact step sizes a bit more (lines 132-134).

We stated the reasons for adding the diffusion term (lines 81-82) and explained this in the appendix (A1 The variational formulation).

**Specific comments:**

The abstract and introduction could flow better. The sentences could be longer or better connected.

We changed the abstract due to additional content and tried to connect the sentences better. We also changed the introduction a bit to get a better flow.

L 4 and 5: I think these sentences can be removed from the abstract "For the step size control, we need a minimization problem. Minimizing a specific convex function is equivalent to solving the full-Stokes equations."

We removed these sentences.

L 18: nonlinear → non-linearly?

We corrected the grammar.

L 18: make it clearer that the stationary variation of these equations is the standard one

We added, how the new shape of glaciers is calculated. This should make clearer why the stationary equations are important (lines 18-20 and in detail in section 6.2).

L 56: These spaces are for linear Stokes, see e.g. the work by Belenki et al: https://www.jstor.org/stable/41582741 for the appropriate spaces

Thanks for the reference. We added an explanation in the appendix (A1 The variational formulation) with reference to this literature (lines 359-360) and explain that the additional diffusion term $\mu_0$ allows us to use the simpler spaces for linear stokes.

L 60: Why do you add a diffusion term? This is non-standard and not included in other ice sheet models, which might make the reader wonder if these results are applicable to their model. I suggest trying to get rid of this term.

We added a short explanation for the diffusion term in the appendix (A1 The variational formulation). Additionally, we did all experiments again with $\mu_0 = 0$ (and added this information in line 85).

However, we partly overwrote our reference solution by our methods that we compared. This led to small mistakes in Fig. 4 and 7 for Newton's method and the Picard iteration with exact step sizes. Now, both methods are a bit better for the later iterations. Additionally, Fig. 3 was slightly wrong for the exact step sizes. Now, the Picard iteration with exact step sizes has a slightly higher difference at the surface to the reference solution and Newton's method with exact step sizes a slightly lower.

Regarding the diffusion term in theory: Without the diffusion term, we have to formulate the problem in a subspace of $W^{1,1+1/n}(\Omega)^N$. For functions $f \in W^{1,1+1/n}(\Omega)^N$, we know

$$\int_\Omega |f|^{1+1/n}\,dx < \infty \quad \text{and} \quad \int_\Omega |\nabla f|^{1+1/n}\,dx < \infty.$$

This is not enough for the well-posedness of the first summand in the Picard iteration and the directional derivative of the variational formulation Eq. (6). The additional diffusion term allows us to have $f \in H^1(\Omega)^N = W^{1,2}(\Omega)^N$. Then, we know

$$\int_\Omega |f|^2\,dx < \infty \quad \text{and} \quad \int_\Omega |\nabla f|^2\,dx < \infty.$$

Now, we can apply Hölder's inequality to show $\int_\Omega |D\boldsymbol{v_{k+1}} : \nabla\boldsymbol{\phi}| < \infty$.

L 71: I would not say that Picard is standard. Include some references for ice sheet models that use Picard rather than Newton. Most probably use a combination of the two.

We added ice models, which use the Picard iteration and replaced "standard" by "common".

L 92-93: It's true that in Hirn $\mu_0$ is 0 (and even delta=0 !) . This should be discussed - why is it not possible to set them to zero in this work?

We changed the simulations to $\mu_0 = 0$. Only for the infinite-dimensional theory $\mu_0 > 0$ is necessary. For the first few iterations, the solution is nearly identical. Some small changes are visible for more iterations.

Algorithm 3: what is the stopping criteria, i.e. how many for-loop iterations do you do?

We used 25 iterations and added this in the text (lines 140-142). One could probably optimize the number of used iterations. However, in terms of the computation time, this is not important.

L 104: For the reader of the cryosphere, I think you should introduce the two step-size control methods with a bit more words, discuss pros and cons etc, rather than just stating the algorithm.

We motivated Armijo and exact step sizes more. Additionally, we introduced the Armijo step sizes first (lines 122-129) and used them as a motivation for the exact step sizes (lines 130-142).

Equation 10-11: Does the fact that the terms are smaller really mean that one can surely say that the solution is not impacted significantly, especially in a time-dependent simulation it is not clear to me.

We set $\mu_0 = 0$ for the simulations. The term $\delta > 0$ is used in ice models. Thus, we did not set $\delta = 0$. However, as we have for typical ranges of $|D\boldsymbol{v}|$ in the computer the relation $D\boldsymbol{v} = D\boldsymbol{v} + \delta$ due to the machine precision, the $\delta$ term should not influence this region. It will have an influence on really small values of $|D\boldsymbol{v}|$, but there are also other inaccuracies due to, e.g., grid resolution, unknown $\Omega$, $B$, or $N$. Thus, we think that a small value of $\delta$ does not change the solution too much compared to the other sources of inaccuracies.

Figure 2: I think it is misrepresentative to compare to models that are not full stokes, please only include full stokes models for the dashed lines.

We specified in section 5.3 that we only compared with full-Stokes models. (*aas2*, *cma1*, *jvj1*, *mmr1*, *rhi3*, and *ssu1*). The displayed standard deviation in Fig. 2 seems to us quite similar to Fig. 6 in [Pattyn2008], just with another scaling on the $y$-axis. However, we forgot the models *aas1*, *oga1*, and *rhi1* and added them calculating for the dashed lines.

About the experiments: I strongly recommend that you add a more complex example, something with more complex geometry and time-dependency. A time-dependent simulation of the arolla glacier maybe.

We added the experiments ISMIP-HOM $E1$ and $E2$ in a time-dependent formulation. For the time-dependent simulation, we had to add

$$- \int_\Omega p \operatorname{div} \boldsymbol{v} \, dx$$

to the functional in Eq. (7) because for the time-dependent problem our initial guesses are not necessary divergence-free.

About the experiments: Add plots with relative error vs CPU-time, to show if the step-size control increases the work per iteration significantly or not

We did not do a relative error vs computation time plot for two main reasons: The computation times for all methods are quite similar for the time-dependent experiments (Table 3 and Table 5). For the three-dimensional example, Newton's method itself takes more time than the Picard iteration (Table 1). However, the step size control is not computationally relevant. Thus, we think there would be too many details in the plots without giving important information. Instead, we added the tables with computation times to give this information. Secondly, one could reduce the computation of the step size control by parallelizing the calculation of the integrals. The *MUMPS* solver automatically uses more processors, if available. Thus, we have a comparision between one optimized algorithm and one proof of concept algorithm. Additionally, one could try if a lower number of bisections for the exact step sizes produces results of similar quality. Therefore, the computational effort for calculating the step sizes could be reduced compared to the complete iteration.

The outlook: This is a little bit informally written.

We rewrote the outlook by adding some possible research directions and reformulating the text a bit more formally.

Additionally, we made minor corrections due to mistakes in the reference in Schmidt, 2023: We removed the statement that the function $G$ would be continuously differentiable and the theoretical statements about the convergence speed. The function $G$ is only Gâteaux differentiable.

---

## Author Comment (AC2)

We thank for taking the time to review the manuscript as the topical editor in great detail. We think that incorporating the critic greatly improves the quality of the manuscript and makes it easier to understand.

All upcoming line and page references refer to the new manuscript.

Dear authors,

Thank you for submitting your manuscript to GMD. Given the challenges we had to find suitable reviewers for your work, I decided to provide a more substantial Topical Editor comment to replace the missing second review.

The paper investigates solutions to the Stokes equations with applications to ice flow modelling. The Stokes equations are discretised using the finite-element method using FENICS. Two solvers implement the Picard and the Newton method, respectively. The authors apply their numerical solver to community benchmarks and discuss convergence rates as function of different step size selection strategies.

Although being an interesting and highly relevant topic, the presented work is insufficient in several aspects, mostly in terms of relevance and quality standards within GMD. In summary, the work seems to be a stripped down version of an already published preprint, where formal mathematical proofs were removed (https://arxiv.org/abs/2307.02930). Also, the work lacks in providing significant geoscientific context with respect to the applications. From the succinct text, it is difficult to grasp the novelty of the proposed work, given that the fact that Newton is faster than Picard is exactly what one would expect for the p-Stokes problem. Although it is a nonlinear problem, the p-Stokes equations are the optimality conditions of a strictly convex functional. As a result, one would expect Newton's method to work very well.

Due to the critic, we added experiments and changed the title a bit. We think that the novelty is the exact step size, which improves the Picard iteration and Newton's method. Especially, the Picard iteration with exact step sizes seems to be a good choice compared to Newton's method as it reliably needs nearly the same number of iterations for every step (Fig. 20, Fig. 22, Table 2 and Table 4).

The presented results are also somewhat strange. The poor convergence of Newton solvers in most of the cases are suspicious and could reflect an implementation issue in the algorithm. It would also be interesting to see on more iteration at which "relative difference" (or error) the Picard solver stalls. Such stalls can in most cases originate from either poor scaling of the error, or an issue with the code. In general, the convergence rate of Newton's method should outperform the Picard rate, which does not seem to be the case here in most cases. Further, the Armijo method should work in a robust way for convex problems like this and it is strange to see such poor convergence. Moreover, very little importance seems to be assigned further exploring these non-expected results.

We added the convergence behavior of Newton's method with Armijo step sizes for different resolutions (Fig. 6). For higher resolutions, the relative difference decreases more. We also added the Picard iteration with Armijo step sizes. This algorithm stalls at nearly the same relative difference as Newton's method with Armijo step sizes. Therefore, the minimum of the convex functional and the solution of the full-Stokes equations seems to be too different for this resolution. The approximation of exact step sizes seem to have less difficulties with lower resolutions.

The non-quadratic convergence rate originates from small $\delta$ values, see the Figure below. [Hirn2013] discussed accuracy problems for small $\delta$ values. However, we think using small $\delta$ values is more suitable for ice models as those also use small $\delta$ values and the best choice of $\delta$ is also problem dependent regarding to [Hirn2013].

The Picard iteration would stall at the relative difference 0 at iteration 80 as the reference solution is the Picard iteration with 80 iterations.

[Figure]

Figure 1: Left: Relative residual norm for different $\delta$ values. Right: Surface velocity for different $\delta$ values. The resolution for solving the experiment is given by 351 grid points in the x-direction and 10 grid points in the y-direction.

Based on this situation, and given the review from the first referee, significant improvement needs to be done to the current manuscript in order to be receivable in GMD. I would suggest, besides following all suggestions and addressing all critics from the first reviewer, you implement the following modifications:

- Extend the introduction to add context, comparison with existing solution strategies in other ice flow models, discuss limitations and better place your work in the glaciological modelling framework.

We added a summary of solution strategies in other ice models using the full-Stokes equations (lines 23-31).

- Check your code implementation as the results you report look suspicious. There is still a bug that prevents quadratic convergence of Newton's solver.

We made some further experiments to discuss the convergence of Newton's method. We changed the resolution: With a higher resolution Newton's method with Armijo step sizes reduces the error more. We think this behavior occurs as we have $J' = G$ in the continuous setting. However, in the discrete setting, we can have $J'_h \neq G_h$. Additionally, the Picard iteration with Armijo step sizes has the same convergence problems as Newton's method with Armijo step sizes (Fig. 4). Newton's method converges faster than the Picard iteration. It does not converge quadratic. We think that this originates from too high values of $\delta$, as [Hirn2013] observed accuracy problems for small $\delta$ values and we added an experiment in [Schmidt2023] with higher $\delta$ values which resulted in faster convergence speed. We can not display the latest version, we submitted to this journal. Instead, we present quite similar figures to the submitted ones in this response, see Fig. 1. On the left plot, we see the relative residual norm for different values of $\delta$. We see that the convergence rate is quadratic for a larger value of $\delta$. On the right plot, we see that for $\delta = 10^{-4}$ the surface velocity is similar to the surface velocity for $\delta = 10^{-12}$. Thus, a larger value of $\delta$ can produce similar results but leads to faster convergence with Newton's method.

- Provide further details about the implementation and performance of the solvers. The convergence plots are for sure interesting, but are not the only results to report from your study.

We added the computation time for each iteration and the step size control for the three-dimensional experiment and the time-dependent experiments (Table 1, 3, and 5).

- Better motivate your various choices at all stages in the manuscript, providing some additional and relevant references in some fields.

We added a motivation for the choice of $\gamma$ in Algorithm 3 (lines 127-128), and for $a$, $b$, and the number of the steps in the for loop in Algorithm 4 (lines 140-142). We also added a motivation for the initial guess (lines 165-167), and the choice of $c$ for the relative local difference (lines 193-194).

- Provide a much more "in-depth" analysis of your results. If no change is observed after checking the code, it may be interesting to compare the behaviour of the solvers on other traditional benchmarks, such as viscous inclusion setup or others.

We think that we explained the behavior with different resolutions, refering to [Hirn2013]. We added the ISMIP-HOM experiments $E1$ and $E2$ with and without time-dependence.

Finally, it would be valuable to know your position regarding the preprint from 2023 which is very similar to this paper and may have already been submitted to a more math-oriented journal.

A new version of the preprint on ArXiv is in review at a more math-oriented journal. However, this manuscript has the following differences:

1. We considered additionally the experiments ISMIP-HOM $A$, $E1$, $E2$, and time-dependent versions of $E1$ and $E2$.

2. The experiments $A$ and $B$ have a local mesh-refinement to reduce the computation time for experiment $A$ and experiment $B$ has the refinement to make it comparable to experiment $A$.

3. We compared the relative difference and the local relative difference for the experiments $A$ and $B$. In the mathematical manuscript, we used the residual norm as our error estimate.

4. This manuscript states the applied algorithms in more details with less mathematical termini. Moreover, the algorithms in the mathematical manuscript are formulated for divergence-free elements. Thus, all pressure terms vanish. In this manuscript, we state all this pressure terms.

5. We added the term $-\int_\Omega p \operatorname{div} \boldsymbol{v}\, dx$ to the functional as this is necessary for time-dependent simulations.

6. The target group for this journal is different to the mathematical manuscript. We aimed at making the key ideas and algorithms better understandable and reusable. Thus, we highlighted the used algorithms.

Taking the time and making the effort to carefully revisit and substantially extend the current work may provide a valuable input for the geoscientific modelling community and could be suited for GMD. However, in the current state, the work seems closer to a rushed submission than a complete paper.

We are convinced that our revision substantially increased the quality of the manuscript and are interested to hear if we could resolve all initial concerns. We are happy to discuss resulting issues and are hopeful to produce a manuscript that is suitable to GMD.

---

## Author Response (AR2)

Dear Ludovic Räss,

Thanks for the fast second review round.

Below are the two review responses to this review round. We hope these responses, the revised manuscript, and the tracked changes answer all remaining concerns.

**Review 1**

Dear authors,

I greatly appreciate the new experiments that you have performed, which I think raises the relevance of this paper. My main concerns are the following:

Thanks for the positive feedback.

- About the summary of how common ice sheet models solves non-linearity: Contrary to what is stated in the new version of the manuscript, I am almost certain that Elmer does have an Armijo step-size control. To give a fair summary I recommend that you contact the developers of these codes.

  Thanks for stating your concerns. We already contacted Fabien Gillet-Chaulet. Fabien confirmed that no Armijo step size control has been implemented. He added that he implemented the strategy to use first a few Picard iterations and then Newton's method for a nonlinear sliding law. We added this information (line 25).

- So you need the artificial viscosity to get a well-posed directional derivative but perform the experiments without an artificial viscosity. Could you expand on how this is possible?

  We think that is a weakness of our mathematical approach: We use the infinite-dimensional function spaces instead of the finite-dimensional finite element spaces. The infinite-dimensional approach is more general as constants do not depend on mesh resolution or used finite elements, but this approach has higher requirements on regularity. In the finite-dimensional analysis, e.g. [Hirn2013], the artificial diffusion term is not necessary. We added this information (lines 165 and 166). Our initial thought was that the artificial diffusion term could improve convergence properties, e.g., convergence speed or stability, for Newton's method. However, we did not observe this.

- Table 4: Are these the right headings?

  We hope that we made the headings for Tables 2 and 4 more clear by writing 'Number of iterations for solving the full-Stokes equations 'in' each time step.' instead of 'before'. We also clarified the headings of Tables 3 and 5.

- Clarify the summary - many readers only read the summary an the abstract

  We clarified the aspects mentioned below. Additionally, we added more information about the method of minimizing a function and determining approximately exact step sizes. We also stated all the experiments that we did shortly. We shifted the last parts, which were an outlook to the outlook section and details about the time measurement of the three-dimensional experiment, to that subsection.

- The first sentence in the summary: We conclude that our simulations are similar to the results of Pattyn et al. Similar in which sense? Make it clear that you mean that the solution is similar.

  We changed the sentence: 'We conclude that our simulations are similar to the results in [Pattyn2008].' to 'We observe that our calculated solutions are similar to those in [Pattyn2008].' (line 345).

- In the summary: "the exact step sizes verify convergence of the Newton method" - i don't understand this sentence.

  We replaced 'verify' with 'ensure' (line 346).

- The discussion on the convergence on Newtons method an its relation to $\delta$ is interesting. Can you expand this? Please summarize the findings of Hirn in more detail.

  We added a paragraph to discuss the experiments from [Hirn2013] in more detail (lines 214-219). However, these experiments are less relevant for glacier simulations, as they have 0 on the right-hand side of the full-Stokes equations instead of $\rho\boldsymbol{g}$.

- A good sanity check could be to try to run another ice sheet model to see how the Newton method converge for different $\delta$.

  We agree that this step is necessary for improving Newton's method in ice sheet models. However, [Hirn2013] stated that a good choice of $\delta$ depends on the nonlinear exponent $N$, the domain $\Omega$, and the mesh size. Moreover, [Hirn2013] did consider Dirichlet zero boundary conditions in the theoretical analysis. The sliding boundary condition could also need another choice of $\delta$. Furthermore, this study considered the residual error compared to the solution. This approach could consider slow-flowing parts of the glacier not enough. Finally, we tried to change $\delta$ in *ISSM*. However, to our understanding, we would have to change every use of $\delta$ in the source code as this variable seems not changeable in the *MATLAB* interface. We also can not just state a value for $\delta$. We would have to make a lot of comparisons between solutions with different domains, boundary conditions, and mesh resolutions. In conclusion, we think that this could be quite interesting, but it is out of the scope of this manuscript. We plan to do this in future work in Angelika's group together with a scientific software engineer, as this requires substantial changes in the foundation of the code structure.

**Review 2**

Dear authors,

Thank you for providing an updated version of your manuscript and for adding the new experiment. The manuscript draft looks in much better shape now, and we could further proceed with minor revisions.

Thanks for the positive feedback.

- Please implement all suggestions and modifications requested or suggested by the reviewer at this stage.

  We added all suggestions as explained in the review response except implementing different values of $\delta$ in a larger ice sheet model. Please see our argumentation for why we did not implement different choices of $\delta$ into an ice sheet model.

- Ideally, it would be very interesting and enlightening to also try to run another ice sheet model to see how the Newton method converge for different $\delta$.

  We agree that this step is necessary for improving Newton's method in ice sheet models. However, [Hirn2013] stated that a good choice of $\delta$ depends on the nonlinear exponent $N$, the domain $\Omega$, and the mesh size. Moreover, [Hirn2013] did consider Dirichlet zero boundary conditions in the theoretical analysis. The sliding boundary condition could also need another choice of $\delta$. Furthermore, this study considered the residual error compared to the solution. This approach could consider slow-flowing parts of the glacier not enough. Finally, we tried to change $\delta$ in *ISSM*. However, to our understanding, we would have to change every use of $\delta$ in the source code as this variable seems not changeable in the *MATLAB* interface. We also can not just state a value for $\delta$. We would have to make a lot of comparisons between

solutions with different domains, boundary conditions, and mesh resolutions. In conclusion, we think that this could be quite interesting, but it is out of the scope of this manuscript. We plan to do this in future work in Angelika's group together with a scientific software engineer, as this requires substantial changes in the foundation of the code structure.

- Please also reach out to the Elmer devs to check what they have implemented and refer to anything relevant in the introduction.

  Thanks for stating your concerns. We already contacted Fabien Gillet-Chaulet. Fabien confirmed that no Armijo step size control has been implemented. He added that he implemented the strategy to use first a few Picard iterations and then Newton's method for a nonlinear sliding law.

- Moreover, I would suggest an alternative title such as "Assessing the benefits of approximately exact step sizes for Picard and Newton solver in simulating ice flow (FEniCS-full-Stokes v.1.3.1)" which better reflects the investigative work you are performing given there are quite some remaining open questions.

  We changed the title. To reflect that there are still open questions, we wrote ' 'We think the' reason for this behavior is that the discretized minimum of the convex functional and the root of the full-Stokes equations are slightly different.' instead of ' 'The' reason for ...' (line 207).

Thank you and best regards, Ludovic Räss

---

## Author Response (AR3)

Dear Ludovic Räss,

Thanks for the possibility to improve the readability.

We did the following for clarification

- We removed the technical terms 'relaxation' and 'functional' and replaced them with 'step size control' and 'function', respectively.

- We added a short discussion about the boundary condition $\boldsymbol{v} \cdot \boldsymbol{n} = 0$ in Sect. 2 to make it clearer that the theory also covers the experiments in Sect. 6.

- We added that the minimum step length for the exact step sizes is 0. Additionally, we made a small mistake in setting up the time-dependent experiment of ISMIP-HOM E1 for the Picard iteration with exact step sizes: We had in Algorithm 4 for the minimum step size $a = 0.5$. We recomputed this experiment setting. As all step sizes for the old run were larger than 1, the results did not change. We only updated the computation times, which were for the complete iteration within the standard deviation. We uploaded the new version to Zenodo and changed the title accordingly.

- The description of the functional $J$ in Eq. (7) was imprecise/wrong. We did add the dependency on the pressure $p$. Thus, we corrected the use of $J$ in Algorithm 3 and Eq. (8). We also added Eq. (9) to explain the exact step sizes.

- We rewrote the discussion of Fig. 20 and Table 2 by discussing both together. Then, we discuss each algorithm only once and not twice within a paragraph. We repeated this procedure with Fig. 22 and Table 4.

- We put the information about the maximum number of iterations in the caption of Fig. 20 and Fig. 22 to make this information easily accessible to readers.

- We discuss both Newton variants after each other and both Picard variants after each other in each experiment. Then, the structure is clearer and we need long technical terms like 'Newton's method with Armijo step sizes with a minimum step size of 0.5' less frequently.

- We replaced the terms 'classical Picard iteration' and 'Picard iteration' with 'Picard iteration without a step size control' to distinct better from the exact step sizes and remove the unprecise formulation 'classical Picard iteration'.

- We asked the copy-editing service from Copernicus Publications. However, they explained that copy-editing would be automatically done after acceptance of the manuscript.

---

## Author Response (AR4)

Dear Ludovic Räss,

Thanks for the helpful and fast review process.

We added the missing city and country for the affiliation.

We also added acknowledgements for the support by DFG within the funding programme Open Access-Publikationskosten.

Best regards
Niko Schmidt